# Learning Human Habits with Rule-Guided Active Inference

**Zhiren Gong**[1,2]*, **Chao Yang**[1], **Wendi Ren**[1], **Shuang Li**[1]†
[1]The Chinese University of Hong Kong, Shenzhen    [2]Nanyang Technological University
zhiren001@e.ntu.edu.sg, 222043011@link.cuhk.edu.cn,
wendiren@link.cuhk.edu.cn, lishuang@cuhk.edu.cn

## Abstract

Humans navigate daily life by combining two modes of behavior: *deliberate planning* in *novel* situations and *fast, automatic responses* in *familiar* ones. Modeling human decision-making therefore requires capturing how people switch between these modes. We present a framework for *learning human habits with rule-guided active inference*, extending the view of the brain as a prediction machine that minimizes mismatches between expectations and observations, and computationally modeling of human(-like) behavior and habits. In our approach, habits emerge as *symbolic rules* that serve as compact, interpretable shortcuts for action. To learn these rules alongside the human models, we design a biologically inspired *wake–sleep algorithm*. In the *wake phase*, the agent engages in active inference on real trajectories: reconstructing states, updating beliefs, and harvesting candidate rules that reliably reduce free energy. In the *sleep phase*, the agent performs generative replay with its world model, refining parameters and consolidating or pruning rules by minimizing joint free energy. This alternating rule–model consolidation lets the agent build a reusable habit library while preserving the flexibility to plan. Experiments on basketball player movements, car-following behavior, medical diagnosis, and visual game strategy demonstrate that our framework improves predictive accuracy and efficiency compared to logic-based, deep learning, LLM-based, model-based RL, and prior active inference baselines, while producing interpretable rules that mirror human-like habits.

## 1 Introduction

Understanding human behavior in complex environments has long been a central goal in both cognitive science (Pylyshyn, 1980) and artificial intelligence (Leichtmann et al., 2023). A large body of work suggests that behavior in humans and other mammals is supported by at least two complementary modes of control: a *goal-directed system* that evaluates actions based on their consequences, and a *habit system* that relies on learned stimulus–response routines, with evidence for partially distinct corticostriatal circuits underlying each (Balleine & O'doherty, 2010; Dolan & Dayan, 2013). In *novel situations*, they engage in *deliberate planning*, drawing on *internal models* of the world to simulate possibilities and anticipate outcomes. In *familiar contexts*, they shift effortlessly into *habitual control*, relying on *rules* or *shortcuts* (Neal et al., 2012) that bypass heavy deliberation and allow *rapid, efficient action*. This smooth interplay between flexible reasoning and automatic habits is a hallmark of *human intelligence*—and capturing it in a *biologically plausible* way remains a key challenge for building models that aspire to human-like adaptability.

Active inference (AIF) (Mazzaglia et al., 2022), a framework rooted in neuroscience and Bayesian principles, offers a *biologically inspired*, *brain-like* account of adaptive behavior. It portrays the person as a *prediction-driven mind* that minimizes *free energy* (Parr & Friston, 2019; Millidge et al., 2021): through *variational free energy* it makes sense of sensations by inferring hidden causes, and through *expected free energy* it *imagines plausible futures*—favoring scenarios that both reduce uncertainty (curiosity-driven understanding) and align with the person's preferences (goal-congruent intentions). In doing so, AIF maintains an *internal generative (world) model* that is continually refined to reduce surprise—providing a principled, unified lens on perception, learning, and action.

---

*Work done as a research assistant at The Chinese University of Hong Kong, Shenzhen.
†Corresponding author.

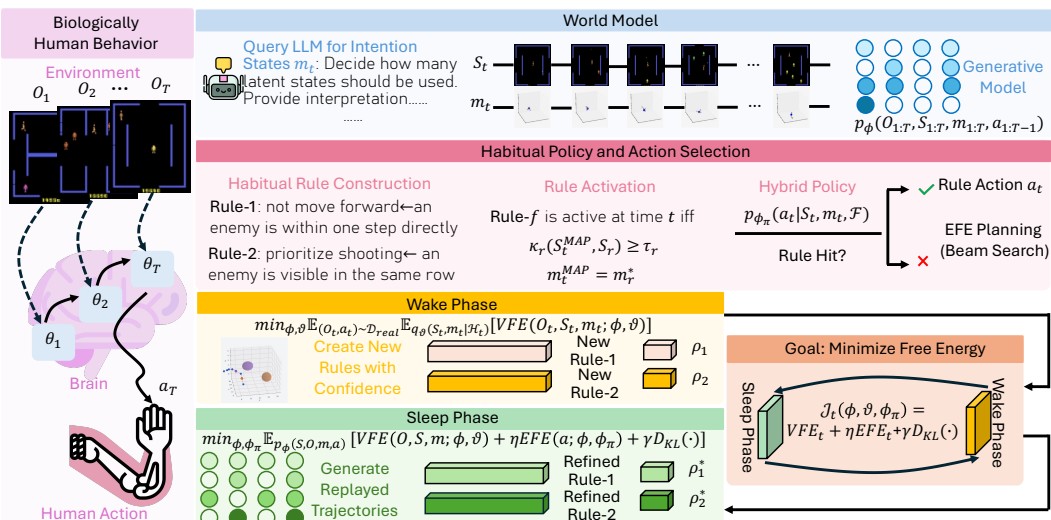

Figure 1: Model framework. "▢": Biologically human behavior, "▢": World model. "▢": Habitual policy and action selection. "▢": Wake phase. "▢": Sleep phase. "▢": Overall goal.

In this work, we use AIF in the control-as-inference sense as a modeling framework for human(-like) behavior (Levine, 2018; Toussaint, 2009), rather than as a new reward-maximizing control agent. *Yet, classical AIF largely operationalizes behavior via prospective planning at each step*, and thus under-specifies three ingredients that are central to human behavior: *habit acquisition*, *habit consolidation*, and *meta-control* over *when to plan* versus *when to act automatically* (Han et al., 2024; Dung Nguyen et al., 2024). Concretely, (*i*) it lacks a mechanism to compress repeated successes into compact, reusable *rules* with confidence; (*ii*) it lacks a principled way to switch modes—using instant, rule-based actions in familiar situations, and calling on costly look-ahead only when uncertainty is high; and (*iii*) it offers no offline process to *consolidate*, *prune*, or *semantically anchor* such rules.

These gaps motivate our approach. We propose a *rule-guided active inference framework* that augments AIF with *habitual policies* learned and refined through a biologically inspired *wake–sleep process* (Hinton et al., 1995; Hewitt et al., 2020; Ellis et al., 2023). Our aim is to use this framework to *fit and explain human (and human-like) action sequences* via a control-as-inference objective. In the *wake phase*, the agent harvests candidate rules from real experience by identifying state–intention–action triples that consistently reduce free energy. In the *sleep phase*, it performs *generative replay* to consolidate, prune, and semantically anchor these rules, so that useful ones are reinforced while spurious ones are discarded. Each rule is grounded in latent state prototypes and interpretable discrete intentions, forming a *neural–symbolic unit* that bridges continuous world models with symbolic decision-making. This hybrid structure enables *instant action* in familiar scenarios through high-confidence rules, while retaining *flexible planning* via expected free energy in novel cases.

Beyond efficiency, the learned rules provide *interpretable structure* that facilitates knowledge transfer and offers insights into the agent's behavior. Altogether, our contributions are threefold: *i)* a biologically inspired extension of active inference tailored to computational modeling of human(-like) habits via rule-guided policies, *ii)* a novel wake–sleep algorithm that jointly learns generative models and symbolic rules under a unified free-energy objective, and *iii)* empirical evidence on human action prediction tasks such as NBA player trajectories, car-following dynamics, medical diagnosis, and visual game strategy, where our framework improves both predictive performance and interpretability compared to deep learning, logic-based, and prior AIF baselines.[1]

## 2 RELATED WORK

**Human Behavior Modeling.** Modeling human behavior is central to applications in public health (Ferguson, 2007; Marsch, 2021), crime analysis (Savage & Vila, 2003), and human–robot collaboration (Dragan & Srinivasa, 2013; Maeda et al., 2017). Probabilistic and deep approaches often focus on predicting dynamics of discrete events: Shen et al. (2018) developed deep models for spatio-temporal events, Zhou et al. (2022) combined neural networks with spatio-temporal point processes, and Chen et al. (2020) proposed continuous-time normalizing flows. These methods

---

[1]Available at `https://github.com/GongZhiren/human-action-active-inference`

achieve strong predictive performance but largely act as black boxes, offering limited insight into the structure of human decisions.

A complementary line emphasizes the role of *habit* in behavior maintenance (Rothman et al., 2009). Examples include HAT (Serra et al., 2018) for cross-task stability, Markovian models of decision-making (Pentland & Liu, 1999), and imitation learning (Ho & Ermon, 2016; Li et al., 2017; Duan et al., 2017) for reproducing expert routines. Yet these approaches encode regularities implicitly, making habits difficult to interpret or arbitrate against planning. Our method addresses this gap by learning compact, interpretable rules within a probabilistic framework that supports both rapid habitual action and flexible planning in novel contexts.

**Logic- and Rule-Based Explanations.** To improve interpretability, recent work incorporates logic rules into predictive models. Li et al. (2021) extract symbolic rules from irregular events, Cao et al. (2023) integrate logic with trajectory models, and Yang et al. (2025) guide temporal point processes with logic priors. Neuro-symbolic methods extend this trend: Li et al. (2023b) impose compositional logic constraints on Transformers for human–object interaction, while Xu et al. (2023) introduce LogicMP for efficient integration of first-order constraints via mean-field inference. d'Avila Garcez et al. (2019) survey neural-symbolic computing, highlighting principled integration of machine learning and reasoning. In hierarchical RL, Bacon et al. (2017) proposed the Option-Critic architecture, which learns options (temporally extended actions) end-to-end, providing a connection between our rule-based controllers and the options literature. While these approaches highlight the value of logic and hierarchical control, rules are often static or imposed post hoc, limiting adaptivity. In contrast, our framework embeds rules directly into the generative process, coupling them with latent states and intentions, optimizing under the free-energy principle, and updating them dynamically through a wake–sleep cycle. This yields interpretable, biologically plausible rules that guide long-horizon planning.

**Active Inference.** Active inference (AIF) offers a unifying framework for perception, action, and learning (Mazzaglia et al., 2022), with applications ranging from psychology (Goette et al., 2023; Demekas et al., 2020) and economics (Henriksen, 2020) to scene construction (Mirza et al., 2016; Heins et al., 2020). Neural extensions have expanded its representational power (Ueltzhöffer, 2018), while subsequent work emphasized action selection via expected free energy (Friston et al., 2016; 2015; 2021; Millidge et al., 2020), efficient objectives (Mazzaglia et al., 2021), and amortized planning through habit networks (Fountas et al., 2020). Tschantz et al. (2020) explored action-oriented representation learning in active inference, focusing on how agents learn representations that support effective action selection. Yet, these approaches largely center on prospective planning, with only ad hoc treatment of habits and no principled mechanism for transitioning between deliberate and routine behavior.

*Our Contribution.* We focus on human behavior modeling and address this gap by embedding symbolic rules directly into the AIF framework. During the wake phase, rules are extracted from experience; during sleep, they are refined via generative replay; and at inference, the inferred active rules guide action selection together with expected-free-energy planning. This captures human(-like) habits as reusable, rule-based shortcuts while retaining a principled planner for novel situations, providing a biologically inspired account of their interaction in human action prediction.

## 3 Background: Sequential Decision-Making via Active Inference

Active inference (AIF) provides a unifying account of *perception*, *learning*, and *action* under a single objective: the minimization of free energy. In sequential decision-making, an agent uses a generative model to explain past observations, update beliefs about hidden states, and plan actions that bring about preferred future outcomes. This dual role naturally leads to two complementary objectives: the *variational free energy* (VFE) for inference and model learning, and the *expected free energy* (EFE) for planning and action selection.

We consider discrete steps $t = 1, \ldots, T$. At each step the agent observes $O_t \in \mathbb{R}^d$, selects an action $a_t \in \mathcal{A}$, and accumulates a history $\mathcal{H}_t = (O_{1:t}, a_{1:t-1})$. Latent states $Z_t \in \mathcal{Z}$ summarize hidden structure relevant for decision-making. A generative model with parameters $\phi$ specifies

$$p_\phi(O_{1:T}, Z_{1:T}, a_{1:T}) = p_\phi(Z_1) \prod_{t=1}^{T} p_\phi(O_t \mid Z_t) \, p_\phi(Z_t \mid Z_{t-1}, a_{t-1}) \, p_\phi(a_t \mid Z_t). \tag{1}$$

**Variational Free Energy (VFE).** Exact inference over $Z_{1:t}$ is intractable, so we introduce an amortized variational posterior $q_\vartheta(Z_t \mid \mathcal{H}_t)$ with parameters $\vartheta$. At each time $\tau$, the per-step

*variational free energy* is defined as
$$\text{VFE}_\tau := \mathbb{E}_{q_\vartheta(Z_\tau \mid \mathcal{H}_\tau)}[-\log p_\phi(O_\tau \mid Z_\tau)] + D_{\text{KL}}\big(q_\vartheta(Z_\tau \mid \mathcal{H}_\tau) \,\|\, p_\phi(Z_\tau \mid Z_{\tau-1}, a_{\tau-1})\big). \tag{2}$$

The first term is a *prediction error*, ensuring latent states explain observations; the second enforces *temporal consistency* with the transition prior. The VFE serves as the learning signal for updating both the generative model parameters $\phi$ and the inference network $\vartheta$.

**Expected free energy (EFE).** In contrast, action selection is prospective. Given a candidate action $a_t$, the agent *rolls out* its generative model over a horizon $H$ to simulate possible futures. We denote the resulting predictive distribution by $q_\phi^{\text{roll}}$, which depends on $\phi$ and the chosen action sequence. At each future step $\tau$, the *expected free energy* is

$$\text{EFE}_{t+\tau}(a) := \mathbb{E}_{q_\phi^{\text{roll}}}\Big[-\log p_{\text{pref}}(O_{t+\tau} \mid Z_{t+\tau}) + D_{\text{KL}}\big(q_\phi^{\text{roll}}(Z_{t+\tau}) \,\|\, p_\phi(Z_{t+\tau} \mid Z_{t+\tau-1}, a_{t+\tau-1})\big)\Big].$$

Here $p_{\text{pref}}$ specifies which future observations are preferred: the agent aims to keep its trajectory within preferred outcomes while maintaining an accurate internal model. The first term measures *risk* (deviation from preferred outcomes), and the second term encourages *epistemic value* by reducing uncertainty about hidden states. We adopt the standard *control-as-inference* view, where preferences are encoded as a biased likelihood over outcomes (Levine, 2018; Toussaint, 2009). Our goal is to *model human behavior from demonstrations*: we assume humans act approximately optimally under some latent preference distribution $p_{\text{pref}}$, but we do not try to recover this distribution (or the underlying reward) explicitly. Instead, we learn the generative model and policy so that the resulting EFE-based controller assigns high likelihood to the observed human actions. In this setting, we use the special case

$$p_{\text{pref}}(O_{t+\tau} \mid Z_{t+\tau}) \propto p_\phi(O_{t+\tau} \mid Z_{t+\tau}),$$

so that preferred futures coincide with high-probability outcomes under the learned observation model, and minimizing EFE favors actions whose predicted outcomes match human behavior. The cumulative expected free energy of action $a$ over horizon $H$ is

$$\text{EFE}_t(a) := \sum_{\tau=1}^{H} \text{EFE}_{t+\tau}(a). \tag{3}$$

Active inference couples these two objectives into an iterative loop. At each time step: (*i*) given new data, the agent *updates its beliefs and generative model* by minimizing (per-step) VFE, aligning latent states with observations; (*ii*) from this belief state, the agent *forecasts future trajectories* under candidate actions, evaluates their cumulative EFE, and executes the *first action* from the trajectory with lowest expected free energy. This integration of retrospective inference and prospective planning defines active inference as a general framework for sequential decision-making.

*Computational Challenges.* Although conceptually elegant, active inference is computationally demanding. Minimizing VFE requires efficient amortized inference for complex latent structures. Minimizing EFE is even more costly, since multi-step rollouts scale rapidly with horizon $H$ and action space size. We will show later that in this paper, we propose to augment AIF with compact *rules* distilled from experience, which can bypass expensive rollouts in *familiar* contexts while preserving full VFE/EFE reasoning in novel ones.

## 4 OUR APPROACH: RULE-GUIDED HABITUAL POLICY

We extend AIF by introducing *compact, latent-grounded rules* that capture habitual responses in familiar contexts. Rules act as symbolic triggers: when recognizable patterns (including the external world patterns and mental states occur in our setting), they prescribe actions directly, yielding fast and interpretable habitual policies. If no rule is triggered (novel scenario), the agent reverts to minimizing EFE through long-horizon rollouts, as in standard AIF. This hybrid arbitration enables seamless switching between fast habits in familiar situations and deliberative planning in novel ones.

Our approach has two main key components: (A) the representation of rules and their integration into a hybrid policy that combines habitual and planning-based control, and (B) a wake–sleep learning algorithm that jointly learn generative model (decoder), recognition network (encoder), and rules under a unified free-energy objective. We will discuss the two components one by one.

**Latent State Representation** First propose to split the (previously generic) latent $Z_t$ into two parts:
$$Z_t = (S_t, m_t),$$

where $S_t \in \mathcal{S}$ denotes the *continuous external world state*, a compact low-dimensional embedding of the environment that supports accurate prediction of observations. In contrast, $m_t \in \{1, \ldots, K\}$ is a *discrete mental state* that encodes intentions, modes, or subgoals.

Given the new latent state representation, we rewrite the generative model (1) as

$$p_\phi(O_{1:T}, S_{1:T}, m_{1:T}, a_{1:T}) = p_\phi(S_1)\, p_\phi(m_1) \prod_{t=1}^{T} p_\phi(O_t \mid S_t)\, p_\phi(S_t \mid S_{t-1}, a_{t-1})$$
$$\times\ p_\phi(m_t \mid m_{t-1}, S_t)\, p_{\phi_\pi}(a_t \mid S_t, m_t). \tag{4}$$

with parameters $\phi$. Here the *world model* links external world states $S_t$ to observations through $p_\phi(O_t \mid S_t)$, while the transition prior $p_\phi(S_t \mid S_{t-1}, a_{t-1})$ captures how the world evolves under actions. The discrete mental state $m_t$ evolves more slowly via $p_\phi(m_t \mid m_{t-1}, S_t)$, providing an interpretable bottleneck of intentions or modes. Finally, the *policy* $p_{\phi_\pi}(a_t \mid S_t, m_t)$ selects actions conditioned jointly on the external state and mental state. Later we will show how this policy can be parameterized with compact symbolic rules, allowing habitual responses to emerge inside the same probabilistic framework.

Given new latent state representation, we approximate the posterior over latent states with an encoder
$$q_\vartheta(S_t, m_t \mid \mathcal{H}_t),$$
which maps the history $\mathcal{H}_t = (O_{1:t}, a_{1:t-1})$ into distributions over both continuous world states and discrete mental states.

## 4.1 (A) Rule Representations and the Hybrid Habitual Policy

Building on the latent split $Z_t = (S_t, m_t)$, we now introduce *symbolic rules* as compact carriers of habitual knowledge.

**Rule Definition.** We define a rule $f$ as an anchored condition–action pair:
$$f:\quad (S_f^\star, m_f^\star) \Rightarrow a_f, \qquad S_f^\star \in \mathcal{S},\ \ m_f^\star \in \{1, \ldots, K\},\ \ a_f \in \mathcal{A},$$
where the continuous anchor $S_f^\star$ encodes a prototype of the external environment, $m_f^\star$ specifies the intention or mode, $(S_f^\star, m_f^\star)$ together describe when the rule becomes active, and $a_f$ is the prescribed action to take. Each rule has a confidence weight $\rho_f \in [0, 1]$, reflecting its reliability. The full rule set is $\mathcal{F} = \{(S_f^\star, m_f^\star, a_f, \rho_f)\}_f$ and is treated as part of the policy parameters $\phi_\pi$. The rule library can be viewed as an amortized mixture over context–action pairs, where each rule defines a prototype component. Our current implementation is an efficient, engineering-driven approximation to full variational learning over $q(m_t)$ and parameters of $p(S_t \mid m_t)$ in a mixture model. A detailed probabilistic view is given in Appendix B.

**Rule Interpretation.** The continuous component $S_f^\star$ summarizes the external world in a compact latent embedding. Although this representation is not directly human-interpretable, its encoded meaning can be probed through the generative world model: by decoding $S_f^\star$ back into observable space through $p_\phi(O_f \mid S_f^\star)$, we can visualize or simulate the prototypical situation it represents. The discrete component $m_f^\star$ is categorical and designed to carry semantic meaning (e.g., *cautious*, *aggressive*, *conserve energy*), often initialized or anchored with interpretable labels.

Together, $(S_f^\star, m_f^\star)$ provide the condition under which a rule applies, yielding policies that are both context-sensitive (through the continuous embedding) and mental-state-driven (through the discrete mode). This design mirrors cognitive science accounts in which habits are grounded jointly in environmental context and internal goals: in familiar scenarios, learned associations trigger rapid action without deliberation.

For instance, a driving agent might acquire the rule brake $\leftarrow (S_f^\star := \text{car ahead very close},\ m_f^\star = \text{cautious})$. Here, the latent world prototype $S_f^\star$ encodes the spatial situation of nearby cars and discrete mode $m_f^\star$ denotes cautious intention. When this familiar combination recurs, the action brake is triggered immediately—bypassing costly rollouts and enabling fast, interpretable habitual control.

**Rule Activation and Recognition of Familiarity.** Given a new history $\mathcal{H}_t$, the encoder produces posterior distributions over latent states. We use MAP estimates for efficient matching:
$$S_t^{\text{MAP}} = \arg\max_s q_\vartheta(S_t = s \mid \mathcal{H}_t), \qquad m_t^{\text{MAP}} = \arg\max_k q_\vartheta(m_t = k \mid S_t, \mathcal{H}_t).$$
A rule $r$ is *active* if both the discrete mode and the continuous context are sufficiently close:
$$\kappa(S_t^{\text{MAP}}, S_r^\star) \geq \tau_r, \qquad m_t^{\text{MAP}} = m_r^\star$$
where $\kappa(\cdot, \cdot)$ is a Gaussian similarity kernel and $\tau_r$ a threshold. Under the Gaussian–mixture view (Appendix B), $\kappa(S_t^{\text{MAP}}, S_r^\star)$ can be interpreted as (proportional to) the posterior responsibility of rule $r$ given a Gaussian prior over $S_t$, and $\tau_r$ simply truncates very small responsibilities. This *soft matching* allows rules to work robustly under noisy data. We adopt MAP estimates instead of sampling because they yield fast, deterministic recognition of familiar situations, consistent with how humans can rapidly "pattern match" to known contexts.

When multiple rules suggest the same action, their contributions are combined by weights:

$$\pi(a \mid S_t^{\text{MAP}}, m_t^{\text{MAP}}) \propto \sum_{r:\, a_r = a} \kappa(S_t^{\text{MAP}}, S_r^{\star}) \, \mathbf{1}\{m_t^{\text{MAP}} = m_r^{\star}\} \, \rho_r,$$

normalized across actions.

**Hybrid Policy.** The final action distribution blends the rule prior with EFE-based planning:

$$p_{\phi_\pi}(a_t \mid S_t, m_t) \propto \pi(a_t \mid S_t^{\text{MAP}}, m_t^{\text{MAP}}) + \big(1 - \mathbf{1}_{\text{rule hit}}\big) \exp\big(-\tau \, \text{EFE}_t(a_t)\big). \quad (5)$$

If reliable rules fire, their prior dominates and habitual actions are executed directly. Otherwise, the agent falls back on deliberative planning via multi-step EFE minimization. The temperature parameter $\tau$ controls how sharply the planning fallback discriminates between actions.

*Connection to Cognition.* This hybrid arbitration is inspired by neuroscientific accounts of the brain's dual systems: habits stored as stimulus–response associations in basal ganglia circuits, and deliberative planning supported by prefrontal and hippocampal structures. This flexible switching between cached habits and on-the-fly planning is a general mammalian capability, consistent with dual-system accounts of dorsomedial vs. dorsolateral striatum and prelimbic vs. infralimbic PFC, as demonstrated in devaluation and contingency-degradation paradigms (Dolan & Dayan, 2013; Cushman & Morris, 2015).

Our framework instantiates this duality: learned rules serve as compact, interpretable "habit circuits" grounded in latent world states and internal modes, while the generative model provides a flexible substrate for foresight and adaptation when novelty arises.

## 4.2 (B) LEARNING ALGORITHM: WAKE–SLEEP

**Joint Objective (Total Free Energy).** We jointly train the generative model $p_\phi$, inference model $q_\vartheta$, and policy parameters $\phi_\pi$ (including rule prototypes) by minimizing a unified *total free-energy objective*:

$$F_t(\phi, \vartheta, \phi_\pi) = \underbrace{\text{VFE}_t(O_t; \phi, \vartheta)}_{\text{fit to observed data}} + \eta \underbrace{\text{EFE}_t(\phi, \phi_\pi)}_{\text{applied to rollouts}} + \gamma \underbrace{D_{\text{KL}}\big(q_\vartheta(m_{t-1} \mid \mathcal{H}_{t-1}) \,\|\, q_\vartheta(m_t \mid \mathcal{H}_t)\big)}_{\text{mental-state consistency}}. \quad (6)$$

Here $\text{VFE}_t(O_t; \phi, \vartheta)$ measures how well $p_\phi$ explains the actual data $O_t$, $\text{EFE}_t(\phi, \phi_\pi)$ is the expected free energy accumulated over a horizon $H$ (with the same form as in planning but used for training on replayed trajectories), and the KL term regularizes discrete mental states. The KL term implements a sticky prior over the discrete mental state, encouraging slow, interpretable mode changes. This can be viewed as arising from a prior $p(m_t \mid m_{t-1})$ that favors persistence, which naturally appears in the VFE expansion when minimizing free energy. The coefficients $\eta, \gamma \geq 0$ balance the contributions, allowing early training to emphasize world-model reconstruction while later phases prioritize accurate reasoning with appropriate regularization.

**Wake Phase (Real Data).** In wake, the agent processes *real trajectories* $\mathcal{D}_{\text{real}}$ and updates $(\phi, \vartheta)$ by minimizing free energy on this dataset:

$$\min_{\phi, \vartheta} \mathbb{E}_{(O_t, a_t) \sim \mathcal{D}_{\text{real}}} \mathbb{E}_{q_\vartheta(S_t, m_t \mid \mathcal{H}_t)} \big[\text{VFE}(O_t, S_t, m_t; \phi, \vartheta)\big].$$

At the same time, we *grow rules* from real data: when a triplet $(S_t^{\text{MAP}}, m_t^{\text{MAP}}, a_t)$ recurs often and yields low free energy, we either

(i) create a new rule $(S_r^{\star}, m_r^{\star}, a_r)$ with initial confidence $\rho_r > 0$, or

(ii) increase the confidence $\rho_r$ of an existing nearby rule.

Continuous anchors are updated as centroids of their assigned latents:

$$S_r^{\star} \leftarrow \frac{\sum_{S \in \mathcal{S}_r^{\text{real}}} w(S)\, S}{\sum_{S \in \mathcal{S}_r^{\text{real}}} w(S)}, \qquad w(S) \propto \exp\big(-\text{VFE}(O, S, m_r^{\star}; \phi, \vartheta)\big).$$

This update can be viewed as an EM-style M-step on the Gaussian means in the mixture view, with $w(S)$ playing the role of (reweighted) responsibilities (see Appendix B).

**Sleep Phase (Replay Data).** In sleep, the agent generates *replayed trajectories* $(S, O, m, a) \sim p_\phi$ and jointly updates $(\phi, \phi_\pi)$ by minimizing

$$\min_{\phi, \phi_\pi} \mathbb{E}_{p_\phi(S, O, m, a)} \big[\text{VFE}(O, S, m; \phi, \vartheta) + \eta \, \text{EFE}(a; \phi, \phi_\pi) + \gamma \, D_{\text{KL}}(\cdot)\big].$$

Here VFE ensures consistency of $p_\phi$ and $q_\vartheta$ under imagination, and EFE provides a training signal for $\phi_\pi$. Rules are *refined* during sleep: centroids $S_r^{\star}$ are updated on replayed latents, confidences $\rho_r$ are adjusted, and low-confidence rules are pruned. Both phases share the same free-energy objective, differing only in their data source. This mirrors human learning, where waking experience updates models and dreaming replay consolidates them.

### 4.3 Overall Algorithm

The overall framework alternates between two coupled processes with details provided in Appendix A:

1. **Learning (wake–sleep):**
   - Wake: update models and grow new rules from real data.
   - Sleep: refine models, consolidate/prune rules, and adjust confidences using replay.

2. **Planning (hybrid policy):** At decision time, if a rule is triggered, the agent acts habitually; otherwise it falls back on model-based search for action selection (e.g., MCTS, $A^\star$, or rollout sampling) to minimize expected free energy.

For efficiency, we first run *blockwise pretraining* to bootstrap the world model by minimizing VFE only (Stage 1), which provides a fast warm-up and stabilizes later training, and then perform *full wake–sleep* cycles with replay using the joint objective in Eq. 6 (Stage 2).

## 5 Experiments

We target cross-domain generality: a shared latent $(S_t, m_t)$ combines a continuous cognitive state $S_t$ with an inner discrete mental state $m_t \in \mathcal{M}$ (intentions/sub-goals) that drives rule triggering and planning. When datasets do not provide salient mental labels, we obtain *LLM-guided* interpretable candidates and select $K$ states via a lightweight matching routine with their detailed prompts and sensitivity study over $K$ deferred to Appendix E and semantic lists deferred to Appendix C.6. Model backbones and training schedules are aligned across domains; full architectural settings and optimizer details appear in Appendix C.7.

### 5.1 Experimental Setup

**Dataset** We evaluate on four domains spanning structured sequences and temporal vision: *(i) NBA SportVU* (Kambhamettu et al., 2024)[2]: $\sim$9.8k train / 2.5k val clips with 7 action classes after LLM-guided feature construction; *(ii) Car-Following* (Li et al., 2023a)[3]: $\sim$19k train / 2.5k val samples with 7 driving modes under an action-centric world model; *(iii) DDXPlus*[4]: $\sim$165k train / 25k val URTI trajectories with 225 actions (ASK/DIAG); *(iv) Atari–Berzerk*[5]: $\sim$16.5k train / 16.5k val grayscale frames with 18 actions. Details of dataset, full preprocessing pipelines, feature construction, and action distributions are all provided in Appendix C.8.

**Baselines** We choose state-of-the-art baselines considering following different fields: *i) Logic based Models*: RNNLogic (Qu et al., 2020) and STLR (Cao et al., 2023). *ii) Deep Neural Models*: Re-Net (Jin et al., 2019), *iii) Active Inference Models*: DAI (Çatal et al., 2020) and DAI-MC (Fountas et al., 2020), *iv) Model-based RL*: DreamerV2 (Hafner et al., 2020)[6], and *v) LLM based Models*: LaTee (Song et al., 2024) and Qwen-0.5B (Team, 2024b) Team (2024a) (a pure LLM baseline that processes observations through a learned encoder and generates action predictions via direct LM).

**Metrics** We evaluate performance along three dimensions: *(i) Accuracy*: (1) Acc@k measures the proportion of correct actions within the next $k$ prediction steps ($k=1, 3, 5$) rather than single-step classification; (2) High-Hit Action Ratio (HHAR) measures accuracy specifically on low-frequency critical actions (e.g., marginal maneuvers in each domain), and is considered satisfactory only if it reaches at least $\sim$80% of the overall Acc. *(ii) Efficiency*: (1) Latency, average inference time per step (ms); (2) Convergence Time (CT), total training time to reach convergence (hours). *(iii) Resource Cost*: Peak Memory (PM), maximum memory usage during training and inference (MB).

### 5.2 Experimental Results and Analysis

Results with best-configuration and baseline comparisons are summarized in Table 1. And the additional experimental results not shown below (especially for Car-Following and DDXPlus)are recorded in the Appendix D.

---

[2] https://github.com/linouk23/NBA-Player-Movements

[3] https://github.com/RomainLITUD/Car-Following-Dataset-HV-vs-AV

[4] https://github.com/mila-iqia/ddxplus

[5] https://zenodo.org/records/3451402

[6] DreamerV2 is adapted as a model-based behavioral predictor: we keep the standard world-model architecture but replace the environment reward with a supervised next-action objective, training from a fixed replay buffer built from human trajectories (offline setting).

Table 1: Overall results under the best configuration (mean±std across 3 random seeds). Acc is reported as Acc@1/3/5; Lat/CT in ms / h.

| Category | Method | NBA SportVU | | Car-Following | |
|---|---|---|---|---|---|
| | | Acc (%) | Lat/CT | Acc (%) | Lat/CT |
| Logic-based | RNNLogic | 67.20/60.55/51.83 (±0.00 / ±0.33 / ±1.50) | **26.90/1.20** (±0.82 / ±0.00) | 72.33/68.14/57.56 (±1.26 / ±1.16 / ±1.45) | **7.58/0.54** (±1.08 / ±0.00) |
| | STLR | 75.25/74.67/70.20 (±0.45 / ±0.72 / ±1.15) | 174.18/3.35 (±0.75 / ±0.33) | 78.90/76.57/75.03 (±1.50 / ±1.82 / ±2.35) | 58.29/1.28 (±0.63 / ±0.20) |
| DeepNN | Re-Net | 72.18/68.45/62.00 (±0.67 / ±0.45 / ±0.67) | 218.42/2.34 (±3.25 / ±0.00) | 76.32/70.71/67.28 (±0.83 / ±1.25 / ±2.06) | 72.23/1.15 (±2.18 / ±0.00) |
| Active Inf. | DAI | 75.36/70.58/62.33 (±1.12 / ±1.48 / ±1.55) | 262.33/1.24 (±4.43 / ±0.01) | 78.86/73.35/68.50 (±0.42 / ±0.28 / ±0.53) | 146.33/0.62 (±1.31 / ±0.01) |
| | DAI-MC | 82.33/80.61/76.47 (±0.87 / ±0.85 / ±1.24) | 386.50/1.52 (±3.62 / ±0.05) | 84.54/82.87/80.25 (±0.34 / ±0.36/ ±0.76) | 189.75/0.79 (±1.55 / ±0.01) |
| LLM-based | LaTee | 78.50/73.32/64.50 (±0.88 / ±1.72 / ±1.06) | 1244.20/4.65 (±10.07 / ±0.90) | 82.36/74.75/71.82 (±1.65/ ±1.12 / ±1.48) | 528.33/2.46 (±8.15 / ±0.76) |
| | Qwen-0.5B | 71.25/64.18/56.42 (±1.12 / ±1.85 / ±1.28) | 2845.35/N/A (±12.45 / —) | 74.85/68.32/62.15 (±1.88 / ±1.45 / ±1.62) | 1256.82/N/A (±9.25 / —) |
| Model-based RL | DreamerV2 | 86.42/83.57/81.65 (±0.47 / ±0.66 / ±0.72) | 52.73/1.75 (±3.25 / ±0.05) | 88.43/85.38/82.33 (±0.54 / ±0.78 / ±1.00) | 38.57/0.92 (±1.04 / ±0.00) |
| **Ours** | | **97.00/91.32/85.69** (±0.51 / ±0.79 / ±0.89) | 35.92/2.59 (±2.78 / ±0.22) | **96.77/95.87/94.16** (±0.34 / ±0.40 / ±0.47) | 10.44/0.65 (±0.59 / ±0.08) |

| Category | Method | DDXPlus (URTI) | | Atari–Berzerk | |
|---|---|---|---|---|---|
| | | Acc (%) | Lat/CT | Acc (%) | Lat/CT |
| Logic-based | RNNLogic | 18.75/16.29/13.28 (±0.00 / ±0.67 / ±1.83) | **124.32/4.36** (±5.71 / ±0.39) | 33.86/27.50/24.38 (±0.34 / ±0.57 / ±0.37) | **72.46/2.04** (±3.27 / ±0.00) |
| | STLR | 22.45/18.33/15.59 (±0.00 / ±0.50 / ±1.18) | 872.00/10.25 (±10.53 / ±0.74) | 45.50/38.72/37.24 (±0.67 / ±1.83/ ±2.25) | 432.35/7.18 (±8.20 / ±0.58) |
| DeepNN | Re-Net | 27.33/20.18/16.17 (±1.32 / ±1.54/ ±1.76) | 1112.42/16.38 (±10.62 / ±0.42) | 40.69/32.48/29.33 (±0.75 / ±0.50/ ±1.84) | 723.02/4.42 (±5.48 / ±0.30) |
| Active Inf. | DAI | 46.82/39.27/34.20 (±0.82 / ±1.24/ ±1.33) | 2033.25/3.88 (±4.59 / ±0.27) | 59.97/52.28/41.46 (±0.72 / ±0.95 / ±1.30) | 977.24/3.67 (±5.51 / ±0.93) |
| | DAI-MC | 57.20/52.15/43.67 (±0.67 / ±1.24/ ±1.49) | 2304.23/4.75 (±6.29 / ±0.47) | 66.82/58.20/48.20 (±0.64 / ±0.77/ ±0.90) | 1429.00/4.95 (±4.87 / ±0.74) |
| LLM-based | LaTee | 28.16/22.14/20.38 (±0.32 / ±0.67/ ±0.92) | 95028.72/20.39 (±45.88 / ±0.82) | 62.18/54.21/49.28 (±0.46 / ±0.68 / ±0.94) | 3230.43/11.64 (±7.61 / ±0.84) |
| | Qwen-0.5B | 24.85/19.62/17.35 (±0.42 / ±0.78 / ±1.05) | 125842.15/N/A (±52.35 / —) | 58.42/51.25/46.18 (±0.58 / ±0.85 / ±1.12) | 4856.72/N/A (±8.75 / —) |
| Model-based RL | DreamerV2 | 64.05/61.48/58.15 (±0.81 / ±0.99 / ±1.03) | 452.25/10.08 (±5.36 / ±0.80) | 76.33/72.18/69.47 (±0.67 / ±1.08 / ±1.33) | 108.02/6.87 (±0.98 / ±0.24) |
| **Ours** | | **79.63/73.58/68.07** (±1.54 / ±2.62 / ±2.60) | 159.45/8.73 (±4.45 / ±0.29) | **85.55/77.20/72.44** (±0.87 / ±0.92 / ±1.05) | 92.63/3.53 (±2.29 / ±0.15) |

**Sequential Dataset** *NBA SportVU.* Figure 2a illustrates: (i) general decreases of $\Delta F$, VFE, EFE, and KL, evidencing steady improvement in fit and decision quality; (ii) rule envelopes in latent space, enabling interpretable reconstructions; (iii) test-set curves (Acc@k, HHAR, latency) with stable convergence and strong accuracy; and (iv) a rule-guided inference case where world-model overlays show rules lowering free energy and improving decisions.

*Car-Following.* This dataset exhibits a small-rule–high-payoff pattern: even compact rule sets saturate accuracy (Acc@3 $\approx$96%) while keeping latency very low ($\approx$10 ms). Training traces show consistent decreases in all free energy components.

*DDXPlus.* On URTI with 225 actions, rule envelopes reliably capture low-frequency edge actions, yielding pronounced gains on HHAR while sustaining strong Acc@k. Rule-triggered choices also reduce per-step inference relative to model-only planning, though absolute latency remains higher due to the large action space.

**Temporal Visual Dataset** *Atari–Berzerk.* With visual inputs (128×128 frames, 18 actions). To further assess generalization, we conducted additional experiments on three *Atari-100k* games (Pong, Breakout, Qbert) with varying complexity levels (see Appendix D.6). Fig. 2b shows that our encoder extracts task-relevant signals structuring $(S_t, m_t)$ accurately with clear explanations after decoder, and rule-guided inference examples demonstrate steady predictive rollouts and consistent decisions.

**Baseline Limitations.** Our baselines face distinct challenges: *(i)* Deep predictive models (Re-Net, Transformer/BiGRU world models) lack rule libraries and explicit habit mechanisms, struggling with rare actions and highly imbalanced action distributions (e.g., DDXPlus's 225 actions with many rare

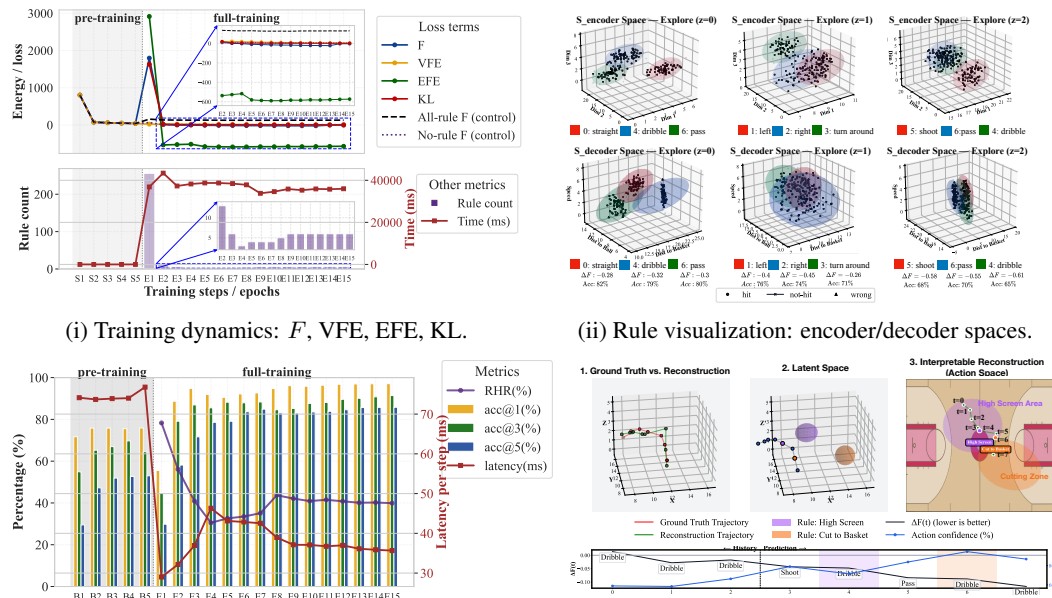

(i) Training dynamics: $F$, VFE, EFE, KL.

(ii) Rule visualization: encoder/decoder spaces.

(iii) Test-set curves: Acc@k, HHAR, Latency.

(iv) Rule-guided inference with world-model.

(a) NBA composite: training dynamics, rule visualization, *test-set* metrics, and inference with world-model overlays. Additional qualitative figures and full curves in other datasets are in the Appendix D.

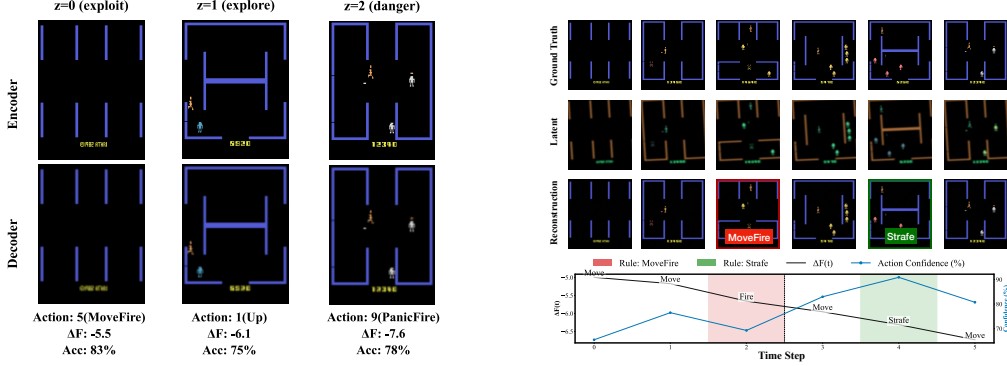

(i) Latent rules and reconstruction (visual scenes).

(ii) Rule-guided inference with world-model overlays.

(b) Atari_Berzerk: rule visualization and inference example, others deferred to the Appendix.

Figure 2: Composites for sequential (NBA) and temporal visual (Atari) datasets.

but critical diagnostic operations). *(ii)* Active inference baselines (DAI/DAI-MC) have only implicit habit policies without explicit latent mental states and symbolic rules, making them inefficient for multimodal behaviors and rare actions, with high computational costs (e.g., DAI-MC latency: 2304ms on DDXPlus). *(iii)* Model-based RL (DreamerV2) is limited by offline settings and sparse rewards, underperforming on pure prediction tasks. *(iv)* Logic-based methods (RNNLogic, STLR) use static or post-hoc extracted rules without joint optimization with world models. *(v)* LLM-based methods (LaTee, Qwen-0.5B) suffer from high latency (e.g., LaTee: 95028ms, Qwen-0.5B: 125842ms on DDXPlus), poor performance (Qwen-0.5B Acc@3: 19.62% vs ours: 73.58%), and lack of generative world models or active inference mechanisms. Our rule-guided active inference addresses these limitations by jointly learning rules and world models under a unified free-energy objective.

## 5.3 RULE–PERFORMANCE TRADE-OFF

Figure 3 shows the Pareto behavior between *accuracy* and *latency* as the rule bank grows. Four consistent patterns emerge:

**(1) Rules speed inference; accuracy is inverted-U in rule size.** As rule count (RC) grows, reference latency drops since cheap rule triggers replace costly planning. Accuracy first rises then falls: compact

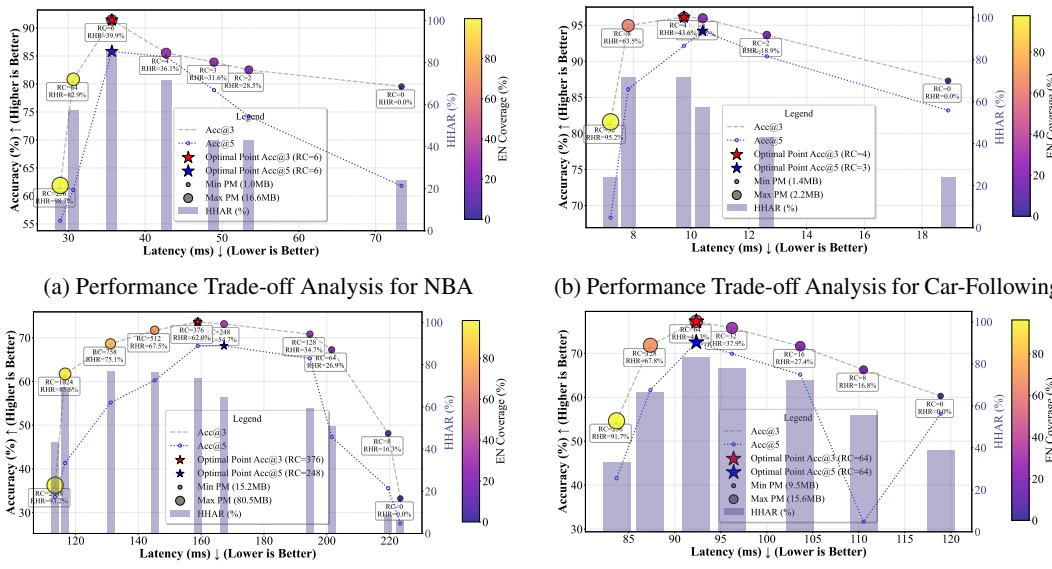

Figure 3: Rule–Performance trade-off across datasets. Each point corresponds to a rule-bank size (RC/EN). Y-axis: accuracy (higher is better); X-axis: latency (lower is better). Bubble size encodes peak memory (PM), color encodes HHAR, and vertical bars denote EN coverage. Stars indicate the Pareto knees used in Table 1.

banks (e.g., NBA at RC≈6, Car-Following at RC≈3–4) capture reliable intents and boost Acc@k, but excessive rules introduce spurious hits and conflicts, degrading decisions despite faster inference.
**(2) Coverage vs. precision diverge after the knee.** RC and rule-hit rate (RHR) keep increasing even as Acc@k declines (HHAR may plateau). Thus *which* rules fire matters more than *how many*: too many rules bias toward noisy or redundant envelopes that compete with the model (e.g., DDXPlus peaks near RC≈376, Atari near RC≈64).
**(3) Memory grows with RC; Pareto knee is optimal.** Peak memory (bubble size) increases with RC. The practical operating point is the knee, balancing accuracy, latency, and memory.
**(4) Rules complement active inference.** EFE-guided planning arbitrates between rollouts and rule triggers via $\Delta F$. With a compact, semantically grounded bank, triggers reduce $\Delta F$ and depth, yielding large latency gains with robust accuracy. Oversized banks cause overlapping envelopes, weakening arbitration and explaining post-peak accuracy drops.

## 5.4 ABLATION STUDY

We ablate four factors relative to the full model: (i) removing rules; (ii) removing the latent intention $m_t$; (iii) dropping generative consistency (VFE, or both VFE and KL); and (iv) greedy rule selection.
**Key findings.** Rules are essential: without them, both accuracy and latency degrade (e.g., NBA Acc@3 91.4→79.5, latency 36→73 ms). Latent intention $m_t$ organizes precision: removing $z$ lowers accuracy and increases latency (e.g., Atari Acc@3 77.3→70.1). Generative consistency is critical: −VFE or Only EFE keeps latency low but causes large accuracy drops (e.g., Car-Following Acc@3 95.9 → 78.5). Greedy rule selection yields the fastest inference (as low as 2.6 ms) but sacrifices accuracy and HHAR, showing instability. Complete ablation study is provided in Appendix D.5.
Therefore, each ablated component removes a distinct capability: speed (no rules), precision (no $z$), cognition grounding (no VFE), or stability (greedy).

## 6 CONCLUSION

We present a cognitive framework that jointly learns a world model and uses it to plan and select future actions via active inference, while a rule engine provides fast, interpretable habitual control. A universal mental-state set enables a single formulation across diverse domains. Experiments on sports tracking, driving, clinical diagnosis, and Atari show strong accuracy under low latency, clear Pareto trade-offs, and rule envelopes that align with human strategies. Overall, the framework captures key aspects of human behavior and substantially enhances interpretability.

REPRODUCIBILITY STATEMENT

We have made extensive efforts to ensure the reproducibility of our results. The complete description of both synthetic dataset generation and real-world dataset preprocessing methods are illustrated in Appendix C. Details of the computational setup, including hardware configuration and software environment, as well as the choice of hyper-parameters are documented in Appendix C.7. We also release our code.

ACKNOWLEDGMENTS

This work was in part supported by the Key Program of the NSFC under grant No. 72495131, the Shenzhen Stability Science Program 2023, the Shenzhen Science and Technology Program No. JCYJ20250604141038013, and Longgang District Key Laboratory of Intelligent Digital Economy Security.

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

APPENDIX OVERVIEW

- **Section A** presents the rule-guided Wake–Sleep framework and online reasoning. It unifies Wake and Sleep into a single pseudocode and details how state reconstruction, belief updating, candidate rule extraction, and model optimization are integrated (Algorithms 1, 2, 3). All definitions (model factorization, VFE/EFE, policy, joint objective) strictly follow the main text.

- **Section B** shows that our model is naturally equivalent to a mixture over latent states, and that rule learning and activation can be interpreted as a lightweight EM-style approximation to inference and parameter updates in this mixture.

- **Section C** describes dataset preprocessing and feature construction for NBA SportVU, Car-Following, DDXPlus, and Atari–Berzerk, including the action spaces and the world-model inputs. It also specifies the semantic interpretations of the discrete internal state $m$ (Section C.6), consolidates key hyperparameters for data/model and optimization/planning/rules (Tables 2–3), and reports action distribution plots across all datasets (Figure 4).

- **Section D** provides additional experimental results beyond the main text: full training dynamics ($\Delta F$, VFE, EFE, KL; Figure 5), testing metrics on held-out sets (Acc@K, rule-hit rate, latency; Figure 6), rule envelopes and visualizations across domains (Figures 7 and 8), end-to-end trajectory visualizations (Figures 9 and 10), full ablation study (Table 4), and Atari-100k supplementary experiment (Section D.6, Table 5).

- **Section E** conducts sensitivity analyses and lists the exact LLM prompts used: NBA action-parameter sensitivity curves (Figure 11), the effect of the latent mental-state cardinality $m$ (Figures 12–14; DDXPlus has fixed $m$), LLM prompts for all datasets (Section E.5), hyperparameter sensitivity analysis (Section E.3, Tables 6–7), and LLM-guided component ablation study (Section E.4).

- **Section F** discusses limitations and broader impact, including domain specificity of mined rules, sensitivity to thresholds and hyperparameters, computational considerations, dependence on world-model quality, and potential directions such as hierarchical timescales and real-world deployment.

## A    WAKE–SLEEP ALGORITHM

---
**Algorithm 1** Rule-Guided Active Inference: Unified Wake–Sleep Cycle

---
**Require:** Dataset $\mathcal{D}$, generative model $p_\phi$, encoder $q_\vartheta$, policy $\phi_\pi$, rule set $\mathcal{F} = \{(S_f^\star, m_f^\star, a_f, \rho_f)\}$
1: Initialize parameters $(\phi, \vartheta, \phi_\pi)$; set $\mathcal{F} \leftarrow \emptyset$
2: **for** each epoch **do**
3:     **Wake phase (real trajectories)**
4:     **for** trajectory $\tau \in \mathcal{D}$ **do**
5:         **for** time $t = 1{:}T$ **do**
6:             Infer $(S_t, m_t) \sim q_\vartheta(S_t, m_t \mid H_t)$
7:             Compute per-step free energy $\text{VFE}_t$ (Eq. 2); accumulate $\Delta F$
8:             **if** $\Delta F < -\delta_F$ recurrently for $(S_t^{\text{MAP}}, m_t^{\text{MAP}}, a_t)$ **then**
9:                 Create or update rule $(S^\star, m^\star, a, \rho)$ with confidence $\rho \leftarrow \rho + \delta_{conf}$
10:                 Update centroid $S^\star \leftarrow \frac{\sum_S w(S)S}{\sum_S w(S)}, w(S) \propto \exp(-\text{VFE}(O, S, m^\star))$
11:             **end if**
12:         **end for**
13:     **end for**
14:     Update $(\phi, \vartheta)$ on real data by minimizing VFE (Eq. 2)

15:     **Sleep phase (replay)**
16:     **for** mini-batch $(S, O, m, a) \sim p_\phi$ **do**
17:         Minimize joint objective $F_t = \text{VFE}_t + \eta \cdot \text{EFE}_t + \gamma \cdot \text{KL}(\cdot)$ (Eq. 6)
18:         Update $(\phi, \phi_\pi)$ by $\nabla F_t$; keep $q_\vartheta$ consistent
19:         Refine $S^\star$, update $\rho$, prune rules with $\rho < \delta_{conf}$
20:     **end for**
21: **end for**

---

This section provides algorithmic details of our framework. We unify the Wake and Sleep phases into a single pseudocode (Algorithm 1), describe the hybrid online reasoning procedure (Algorithm 2),

and summarize the two-stage training schedule (Algorithm 3). All definitions and objectives strictly follow the main text: the generative model factorization (Eq. 1), the variational free energy (VFE, Eq. 2), the expected free energy (EFE, Eq. 3), the rule fusion policy (Eq. 5), and the joint objective (Eq. 6).

---

**Algorithm 2** Online Reasoning with Rule Fusion and Planning

---

**Require:** History $H_t$, encoder $q_\vartheta$, model $p_\phi$, rule set $\mathcal{F}$, horizon $H$, beam width $K$
1: Infer $(S_t, m_t) \sim q_\vartheta(S_t, m_t \mid H_t)$
2: **Rule prior:** for each $r \in \mathcal{F}$, if $\kappa(S_t, S_r^\star) \geq \tau_r$ and $m_t = m_r^\star$, add weighted vote $\rho_r$ for $a_r$
3: If rules triggered, form $\pi_{\text{rule}}(a \mid S_t, m_t)$; else set $\pi_{\text{rule}} \equiv 0$
4: **Planning fallback:** evaluate candidate actions by EFE (Eq. 3) via beam search (width $K$, horizon $H$)
5: Fuse distributions: $p_{\phi_\pi}(a_t \mid S_t, m_t, \mathcal{F}) \propto \pi_{\text{rule}}(a_t) + (1 - \mathbf{1}_{\text{rule hit}}) \exp\{-\eta \cdot \text{EFE}_t(a_t)\}$ (Eq. 5)

---

### A.1 WAKE & SLEEP PHASES

**Wake.** Operates on real trajectories: latent inference by $q_\vartheta$, per-step free energy from Eq. 2, and candidate rules from recurring $(S, m, a)$ triplets that stably reduce free energy. Rule anchors are updated as weighted centroids ($\exp(-\text{VFE})$ as weights).

**Sleep.** Uses replay samples from $p_\phi$, minimizing the composite loss $J_t$ (Eq. (5)). Rules are refined (anchor shift, confidence update) and pruned if redundant or low-confidence.

This division follows the classic wake–sleep paradigm (Hinton et al., 1995; Friston et al., 2015) but adapted to rule-guided active inference.

### A.2 TRAINING SCHEDULE

We adopt a two-stage schedule to stabilize learning.

---

**Algorithm 3** Two-Stage Training: Blockwise Pretraining and Full Wake–Sleep

---

**Require:** Dataset $\mathcal{D}$, blocks $\{\mathcal{D}_b\}_{b=1}^B$
1: **Stage 1: Blockwise pretraining**
2: **for** block $b = 1{:}B$ **do**
3:     **for** mini-batch $(O, a) \in \mathcal{D}_b$ **do**
4:         Infer $S \sim q_\vartheta(S \mid O)$; reconstruct $O$
5:         Update $(\phi, \vartheta)$ by minimizing VFE only (Eq. 2)
6:     **end for**
7: **end for**
8: **Stage 2: Full Wake–Sleep training**
9: **for** epoch $= 1{:}E$ **do**
10:     Run Wake phase on real data (Alg. 1, lines 3–12)
11:     Run Sleep phase with replay (Alg. 1, lines 14–18)
12:     Refine/prune rule pool; update confidences
13: **end for**

---

### A.3 SUMMARY

- Wake extracts rules via $\Delta F$-based improvements and updates confidences. - Sleep uses replay to refine rules and jointly minimize $J_t$ (Eq. 6). - Online reasoning integrates rule priors and planning (Eq. 5), realized via beam search or MCTS. - Training proceeds in two stages: blockwise pretraining for fast bootstrapping, then full Wake–Sleep for convergence.

## B MIXTURE MODEL OVER LATENT STATES AND CONNECTION TO EM-STYLE UPDATES

In the main text (Sec. 4.1), we introduce rules as compact, latent-grounded carriers of habitual knowledge. Here we make explicit a probabilistic interpretation of the rule library as a *Gaussian mixture* over latent contexts, and clarify how our rule scores and updates correspond to approximate EM steps.

**Gaussian mixture over latent contexts.** Recall that the latent state factorizes as $Z_t = (S_t, m_t)$, with continuous $S_t$ and discrete $m_t \in \{1, \ldots, K\}$. Each rule $r$ stores a prototype
$$(S_r^\star, m_r^\star, a_r, \rho_r),$$

where $S_r^\star$ is a continuous anchor in latent state space, $m_r^\star$ is a discrete mental-state label, $a_r$ is the associated action, and $\rho_r \in [0, 1]$ is a nonnegative rule weight. The full rule set

$$\mathcal{F} = \{(S_r^\star, m_r^\star, a_r, \rho_r)\}_r$$

can be viewed as the parameters of a Gaussian mixture over latent contexts.

Introduce a latent rule index $r_t \in \{1, \ldots, |\mathcal{F}|\}$ at each time step, with categorical prior

$$p(r_t = r) = \pi_r, \qquad \sum_r \pi_r = 1,$$

and component-specific priors over $(S_t, m_t)$ of the form

$$p(S_t \mid r_t = r) = \mathcal{N}(S_t; \mu_r, \Sigma_r), \qquad p(m_t \mid r_t = r) = \delta(m_t = m_r^\star),$$

where $\mathcal{N}(\cdot; \mu_r, \Sigma_r)$ is a Gaussian over the continuous latent state and $\delta(\cdot)$ is a Kronecker delta fixing the discrete mental state. In our implementation, we tie the mean to the stored anchor, $\mu_r = S_r^\star$, and use a shared covariance $\Sigma_r = \sigma^2 I$, so

$$p(S_t \mid r_t = r) = \mathcal{N}(S_t; S_r^\star, \sigma^2 I) \propto \exp\big(-\tfrac{1}{2\sigma^2}\|S_t - S_r^\star\|^2\big).$$

Under this mixture, the posterior over rules factorizes as

$$q(r_t = r \mid S_t, m_t) \;\propto\; p(r_t = r)\, p(S_t \mid r_t = r)\, p(m_t \mid r_t = r) \tag{7}$$
$$\propto\; \pi_r\, \mathcal{N}(S_t; S_r^\star, \sigma^2 I)\, \mathbf{1}\{m_t = m_r^\star\}.$$

Thus the posterior responsibility of rule $r$ is a *Gaussian function* of the distance between $S_t$ and the anchor $S_r^\star$, modulated by whether $m_t$ matches $m_r^\star$.

**Connection to the encoder and rule scores.** In the main text, given a new history $\mathcal{H}_t$, the encoder $q_\vartheta(Z_t \mid \mathcal{H}_t)$ produces a posterior over $(S_t, m_t)$, and we use its MAP estimate

$$S_t^{\mathrm{MAP}} = \arg\max_s q_\vartheta(S_t = s \mid \mathcal{H}_t), \qquad m_t^{\mathrm{MAP}} = \arg\max_k q_\vartheta(m_t = k \mid S_t, \mathcal{H}_t).$$

Assuming $q_\vartheta(S_t \mid \mathcal{H}_t)$ is approximately Gaussian with mean $S_t^{\mathrm{MAP}}$, combining this approximate posterior with the Gaussian prior $p(S_t \mid r_t = r)$ yields log-responsibilities that are quadratic in $\|S_t^{\mathrm{MAP}} - S_r^\star\|$, i.e. Gaussian in distance.

We instantiate this by defining Gaussian kernel scores

$$w_r(t) \;=\; \kappa(S_t^{\mathrm{MAP}}, S_r^\star)\, \mathbf{1}\{m_t^{\mathrm{MAP}} = m_r^\star\}, \qquad \kappa(S_t^{\mathrm{MAP}}, S_r^\star) := \exp\Big(-\tfrac{1}{2\sigma^2}\|S_t^{\mathrm{MAP}} - S_r^\star\|^2\Big).$$

These scores act as unnormalized responsibilities of rule $r$ for the current context. Comparing with the mixture posterior above, this corresponds to the approximation

$$q(r_t = r \mid S_t^{\mathrm{MAP}}, m_t^{\mathrm{MAP}}) \;\propto\; \pi_r\, \exp\Big(-\tfrac{1}{2\sigma^2}\|S_t^{\mathrm{MAP}} - S_r^\star\|^2\Big)\, \mathbf{1}\{m_t^{\mathrm{MAP}} = m_r^\star\},$$

i.e. a Gaussian posterior over the rule index as a function of the latent state. In our implementation, we absorb $\pi_r$ into the learned rule weight $\rho_r$, and treat $\kappa(\cdot, \cdot)$ as the dominant distance-based term. A rule is declared *active* if

$$\max_r w_r(t) \;\geq\; \tau_r,$$

so the threshold $\tau_r$ simply truncates very small posterior responsibilities.

When multiple rules suggest the same action $a$, we define a rule-induced action distribution

$$\pi(a \mid S_t^{\mathrm{MAP}}, m_t^{\mathrm{MAP}}) \;\propto\; \sum_{r:\, a_r = a} w_r(t)\, \rho_r,$$

normalized over $a$. Under the Gaussian mixture view, $w_r(t)$ approximates the context-dependent responsibility $q(r_t = r \mid S_t^{\mathrm{MAP}}, m_t^{\mathrm{MAP}})$, and $\rho_r$ is the learned mixture weight of rule $r$.

**Connection to EM-style updates.** In the wake phase, we grow and refine rules from real trajectories. When a triplet $(S_t^{\mathrm{MAP}}, m_t^{\mathrm{MAP}}, a_t)$ recurs often with low free energy, we either (i) create a new rule $(S_r^\star, m_r^\star, a_r)$ with initial weight $\rho_r > 0$, or (ii) increase the weight $\rho_r$ of an existing nearby rule. Continuous anchors are updated as weighted Gaussian centroids:

$$S_r^\star \;\leftarrow\; \frac{\sum_{S \in \mathcal{S}_r^{\mathrm{real}}} u(S)\, S}{\sum_{S \in \mathcal{S}_r^{\mathrm{real}}} u(S)}, \qquad u(S) \propto \exp\big(-\mathrm{VFE}(O, S, m_r^\star; \phi, \vartheta)\big),$$

where $\mathcal{S}_r^{\mathrm{real}}$ collects latent states assigned to rule $r$ on real data. This update is an EM-style M-step on the Gaussian means $\mu_r$ in the mixture model, with $u(S)$ playing the role of (reweighted) responsibilities.

During sleep, rules are further refined using replayed trajectories: anchors $S_r^\star$ are updated on replayed latents, rule weights $\rho_r$ are adjusted based on how often and how well they explain latent contexts, and rules with persistently low effective weight are pruned. Birth (when no rule's responsibility

exceeds the threshold) and pruning (when a rule receives negligible responsibility) play the role of cluster creation/deletion in nonparametric Gaussian mixture models (e.g., DP-means).

Although we present the rule mechanism in algorithmic terms (kernels, thresholds, centroids), it admits a natural Bayesian interpretation: rules correspond to Gaussian mixture components over latent contexts $(S_t, m_t)$, with Gaussian priors over $S_t$ and fixed labels for $m_t$; $w_r(t)$ approximate posterior responsibilities over the rule index; and the anchor and weight updates correspond to EM-style M-steps. This interpretation clarifies that the rule library is not an ad-hoc heuristic, but a structured, *lightweight and computationally efficient* approximation to Gaussian mixture modeling in latent AIF space.

## C  DATASET PREPROCESSING AND FEATURE CONSTRUCTION

### C.1  DATASET

**NBA SportVU.** We extract frame-level coordinates of the ball and ten players from SportVU event data, filtering invalid samples (e.g., missing entities or frames with fewer than 11 tracked objects) and sampling up to 20 clips per game. Since raw coordinates are insufficient for rule construction, we construct new parameterized features under the guidance of large language models (LLMs) and conduct a rationality analysis to validate feature choices. This yields interpretable features such as relative velocity, spacing, and formation compactness. The final dataset contains about 9.8k training samples and 2.5k validation samples, with an action space of 7 classes (e.g., straight run, turn, dribble).

**Car-Following.** The original traffic data do not explicitly provide environmental dynamics. We therefore define an *action world model*, where observation features are constructed from action sampling statistics (e.g., acceleration and headway patterns) to characterize driving dynamics. The resulting dataset consists of about 19k training samples and 2.5k validation samples, with 7 driving modes such as cruise, follow, and accelerate.

**DDXPlus.** The DDXPlus dataset consists of diagnostic trajectories generated by a multi-disease Naive Bayes teacher model. We select URTI, the most frequent disease, as the target condition for diagnostic inference. Each trajectory contains a sequence of ASK actions (doctor's inquiries) followed by a final DIAG action. After preprocessing and formatting, we obtain about 165k training samples and 25k validation samples, with an action vocabulary of 225 classes.

**Atari–Berzerk.** We use high-score human demonstration trajectories in the Atari *Berzerk* game (representing strong human intelligence). Raw RGB frames are converted into $128 \times 128$ grayscale images. Each frame is paired with the corresponding human action, drawn from 18 discrete classes (movement, positioning, firing, etc.). The processed dataset contains about 16.5k training samples and 16.5k validation samples.

### C.2  NBA SPORTVU

**Action space.** We follow the analytic definitions and symbolic feature construction described in the main text and experimental log. Seven discrete basketball actions are defined from player–ball relations. Let $p_t \in \mathbb{R}^2$ denote the player's position, $b_t \in \mathbb{R}^2$ the ball position, and $h_t \in \mathbb{R}^2$ the unit heading vector ($h_t = \frac{p_t - p_{t-1}}{\|p_t - p_{t-1}\|}$). Define constants

$$D_{\text{dribble}} = 2 \text{ ft}, \quad D_{\text{release}} = 6 \text{ ft}, \quad D_{\text{receive}} = 2 \text{ ft}.$$

Then the discrete action $a_t$ is given by

$$a_t = \begin{cases} 4, & \|p_t - b_t\| \leq D_{\text{dribble}}, \\ 5, & \|p_{t-1} - b_{t-1}\| \leq D_{\text{dribble}}, \|p_t - b_t\| \geq D_{\text{release}}, \min_j \|y_t^j - b_t\| \leq D_{\text{receive}}, \\ 6, & \|p_{t-1} - b_{t-1}\| \leq D_{\text{dribble}}, \|p_t - b_t\| \geq D_{\text{release}}, \min_j \|y_t^j - b_t\| > D_{\text{receive}}, \\ 3, & (h_{t-1} \cdot h_t) > \epsilon, \\ 2, & (h_{t-1} \cdot h_t) < -\epsilon, \\ 1, & \text{otherwise if } (h_{t-1} \cdot h_t) < 0, \\ 0, & \text{otherwise if } (h_{t-1} \cdot h_t) \geq 0, \end{cases}$$

where $\epsilon = \pi/18$ (10°) and $y_t^j$ are defender positions.

**Symbolic feature extraction.** For each historical frame $t = 1, \ldots, H$:

**(i) Soft direction–kernel features.** For opponent $j$ and basis direction $b_i \in \{(1,0), (0,1), (-1,0), (0,-1)\}$, define

$$\phi_{t,j,i}^{\text{dir}} = \exp\left(-\frac{\|p_t - y_t^j\|^2}{2\sigma^2}\right) \cdot \max\left(b_i^\top u_t^j, 0\right),$$

where $u_t^j = \frac{y_t^j - p_t}{\|y_t^j - p_t\|}$ and $\sigma = 10$.

**(ii) Relational distance/angle features.** Let $d_j = \|p_t - y_t^j\|$ and sort $d_{(1)} \leq d_{(2)} \leq d_{(3)}$. Define

$$d_{\text{rim}} = \|p_t - r\|, \quad \theta_{\text{rim}} = \arctan 2(r_y - p_y, r_x - p_x), \quad d_{\text{mean}} = \frac{1}{|\mathcal{O}| - 1} \sum_{j \in \mathcal{O} \setminus \{\text{handler}\}} d_j.$$

Then form

$$\phi_t^{\text{rel}} = [\, d_{(1)}, \ d_{(2)}, \ d_{(3)}, \ d_{\text{rim}}, \ \theta_{\text{rim}}, \ d_{\text{mean}} \,].$$

Further sensitivity analysis of LLM-guided action parameters, as well as the prompts used for generating symbolic actions, are provided in Appendix E.

## C.3 CAR-FOLLOWING

For the car-following domain, we use the open-source trajectory dataset where each run is recorded as a sequence of driving regimes. Data are extracted from HDF5 files and organized into fixed-length training samples.

**Preprocessing.** Each sample is represented by:
- Previous-$K$ one-hot encoded regimes (history of executed actions).
- $dt$: time interval between consecutive frames.
- $run\_len$: cumulative length of the current driving run.
- $since\_last$: time elapsed since the last regime change.

This yields a structured observation vector per frame.

**Action space.** We adopt seven discrete regimes (e.g., constant speed, acceleration, deceleration, free-flow, car-following, closing-in, and emergency braking), directly encoded in one-hot form.

## C.4 DDXPLUS

We use the DDXPlus dataset, which consists of synthetic diagnostic dialogues covering multiple pathologies. Each trajectory is represented as a sequence of evidence acquisition (ASK) and diagnostic (DIAG) actions.

**Preprocessing.** Each record in the original dataset contains a set of evidences with associated ground-truth diagnoses. We construct training samples as:
- Evidence parsing: convert raw evidences into tokenized observations.
- ASK/DIAG sequence construction: generate trajectories where the agent sequentially asks for evidence or outputs a diagnosis.
- Subset selection: restrict to URTI pathology for controlled experiments, with limits on train/val/test sizes as documented in the experimental log.

**World model representation.** For each evidence $e$, we compute an embedding E2V($e$) (Evidence2Vec). At each step, the state representation concatenates:
- The Evidence2Vec embedding of the most recent evidence.
- A Top-$K$ posterior vector over candidate diagnoses.
- The entropy of the posterior distribution as an uncertainty measure.

**Action space.** The action vocabulary consists of all ASK tokens (corresponding to medical evidences) and DIAG tokens (candidate diagnoses). This yields a discrete action set comparable to multi-class classification.

## C.5 ATARI-BERZERK

We use human gameplay trajectories on the Atari BERZERK environment, where each frame is a raw image and actions correspond to discrete joystick commands.

**Preprocessing.** Game episodes are unpacked into frame sequences. Each frame is preprocessed by:
- Resizing to $128 \times 128$ pixels.
- Converting to grayscale.
- Normalizing pixel intensities to $[0, 1]$.

Sequences are then segmented into fixed horizons with stride, producing training samples aligned with action labels.

**World model representation.** A vision encoder–decoder architecture is used to reconstruct frames and predict latent states. The encoder extracts spatial features, while the decoder ensures faithful reconstruction for VFE minimization. Temporal dependencies are modeled by a Transformer-based dynamics module.

**Action space.** We adopt the original Atari action set with 18 discrete joystick commands (e.g., move directions, fire, stay). Each action is treated as a one-hot token in training.

## C.6 Internal State Definitions

**LLM-guided mental-state matching.** We construct a uniform matching method for mental states $\mathcal{M} = \{m_1, \ldots, m_K\}$, which correspond to interpretable intentions or sub-goals and, together with the external continuous state $S_t$, form the latent pair $(S_t, m_t)$ that drives rule triggering and planning. Unless a dataset already provides salient labels that can directly play the role of mental states (e.g., severity judgments in DDXPlus), we rely on large language models (LLMs) as expert guidance to generate discrete candidate mental states conditioned on task-specific context, yielding semantically grounded labels (e.g., defensive/offensive sub-goals in sports, conservative/aggressive modes in driving). The exact prompts and semantic candidate lists are given in Appendix E.5, while a sensitivity analysis over the number of mental states $K$ appears in Appendix E.2.

At the optimal number of states, we interpret each $m$ as follows:

**NBA SportVU.** Three mental states are used:
- $m = 0$: Habitual/exploit — stable ball handling or passing routines.
- $m = 1$: Explore — probing maneuvers or less frequent moves.
- $m = 2$: Subgoal switching — transitions between attack patterns (e.g., dribble $\rightarrow$ shoot).

**Car-Following.** Two mental states are used:
- $m = 0$: Exploit — stable regimes such as constant speed or smooth following.
- $m = 1$: Explore — rare or abrupt switching regimes (e.g., sudden braking, acceleration).

**DDXPlus.** Five mental states are used:
- $m = 1$: Early exploration of evidences.
- $m = 2$: Focused questioning around relevant symptoms.
- $m = 3$: Transition phase toward diagnosis.
- $m = 4$: Confident diagnosis with supporting evidence.
- $m = 5$: Over-exploration or redundant questioning.

**Atari-Berzerk.** Four mental states are used:
- $m = 0$: Exploit — regular movement in safe zones.
- $m = 1$: Explore — irregular actions or novel paths.
- $m = 2$: Danger/Escape — evasive maneuvers when surrounded by enemies.
- $m = 3$: Aggressive attack — high-risk firing at opponents.

These semantic interpretations are derived from LLM-guided prompts and verified by sensitivity analysis of the number of latent states (Appendix E.2).

## C.7 Training Parameters

**Model backbones across domains.** For structured sequential data (NBA, Car-Following, DDXPlus), the generative model is parameterized by a Transformer (NBA, DDXPlus) or a Gaussian BiGRU (Car-Following), modeling latent dynamics $p_\phi(S_t \mid S_{t-1}, a_{t-1})$ and observation reconstruction $p_\phi(O_t \mid S_t)$. The inference model is a corresponding Transformer/BiGRU encoder approximating $q_\vartheta(S_t, m_t \mid H_t)$. The policy head conditions on $(S_t, m_t)$ and outputs discrete action distributions via a fully connected layer with softmax or Gumbel-softmax. For temporal visual data (Atari–Berzerk),

Table 2: Data windows, sizes, action vocab, model components, and parameter counts.

| | NBA SportVU | Car-Following |
|---|---|---|
| Hist / Pred window | 10 / 5 | 10 / 5 |
| Train / Val / Test | 9,840 / 2,460 / — | 19,250 / 2,500 / — |
| #Actions | 7 | 7 |
| Batch | 64 | 64 |
| Input resolution | — | — |
| Model components | Transformer | BiGRU |
| Model input features | obs_hist / obs_pred | prevK, $dt$, run_len, since_last |
| #Params ($\times 10^6$) | ∼3.5M | ∼2.2M |
| | **DDXPlus** | **Atari–Berzerk** |
| Hist / Pred window | all / 5 | 10 / 5 |
| Train / Val / Test | ≈165k / 25k / 25k      (shard 5k) | 16,541 / 16,581 / — |
| #Actions | 225 (ASK+DIAG) | 18 |
| Batch | 64 | 64 |
| Input resolution | — | 128×128 (gray) |
| Model components | Evidence2Vec + Transformer | CNN encoder–decoder + Transformer |
| Model input features | symptom/test embeddings (Evidence2Vec) | raw frames |
| #Params ($\times 10^6$) | ∼8.5M | ∼13.6M |

Table 3: Optimization schedule, planning, and rule-related hyperparameters.

| | NBA SportVU | Car-Following | DDXPlus | Atari–Berzerk |
|---|---|---|---|---|
| Pretrain blocks / epochs | 5 / 1 | 5 / 1 | 5 / 1 | 5 / 1 |
| Pretrain LR | $1\times10^{-3}$ | $1\times10^{-4}$ | $1\times10^{-3}$ | $1\times10^{-4}$ |
| Full epochs | 15 | 15 | 15 | 15 |
| Full LR | $3\times10^{-4}$ | $1\times10^{-4}$ | $3\times10^{-4}$ | $1\times10^{-4}$ |
| Weight decay | 0.01 | 0.01 | 0.01 | 0.01 |
| Warmup steps | 2000 | 4000 | 16000 | 2000 |
| $(\eta, \gamma)$ | (0.05, 1) | (0.01, 1) | (0.01, 10) | (0.2, 5) |
| Planning horizon | 4 | 4 | 4 | 4 |
| Beam width $k$ | 6 | 6 | 6 | 6 |
| Rule thresholds $(\Delta F, \delta_{sup}, \delta_{conf})$ | (0.25, 0.75, 0.75) | (0.25, 0.75, 0.75) | (0.5, —, 0.75) | (5, 0.75, 0.75) |

the generative model uses a convolutional encoder (CNN) to extract frame-level features, combined with a temporal Transformer to model latent dynamics, and a decoder to reconstruct $p_\phi(O_t \mid S_t)$, while the inference model jointly processes CNN features with temporal modules to approximate $q_\vartheta(S_t, m_t \mid H_t)$. The policy head applies an MLP with softmax on the latent representation to output distributions over actions.

Tables 2 and 3 list dataset-specific settings strictly taken from the experimental log and consistent with the main-text modules.

**Notes.** "—" indicates not specified in the log; we keep it unspecified here. Rule thresholds are dataset-specific as recorded. Planning uses beam search; horizon refers to the planning horizon (not the hist/pred window). All settings align with Appendix A.

## C.8 ACTION DISTRIBUTION

To further illustrate the characteristics of our datasets, we plot the empirical action distributions for all domains (NBA, Car-Following, DDXPlus, Atari–Berzerk) in Figure 4. These histograms reveal highly imbalanced patterns: for instance, NBA is dominated by the `straight` action (∼69%), Car-Following by cruising (F, ∼55%), DDXPlus by a few high-frequency ASK/DIAG queries (top-10 actions > 60%), while Atari–Berzerk is dominated by move-and-fire combinations (> 50%) with many rare actions < 5%.

Despite this imbalance, our framework adapts well: frequent actions are mainly handled by active inference and multi-step planning, while infrequent actions are captured effectively by mined rules.

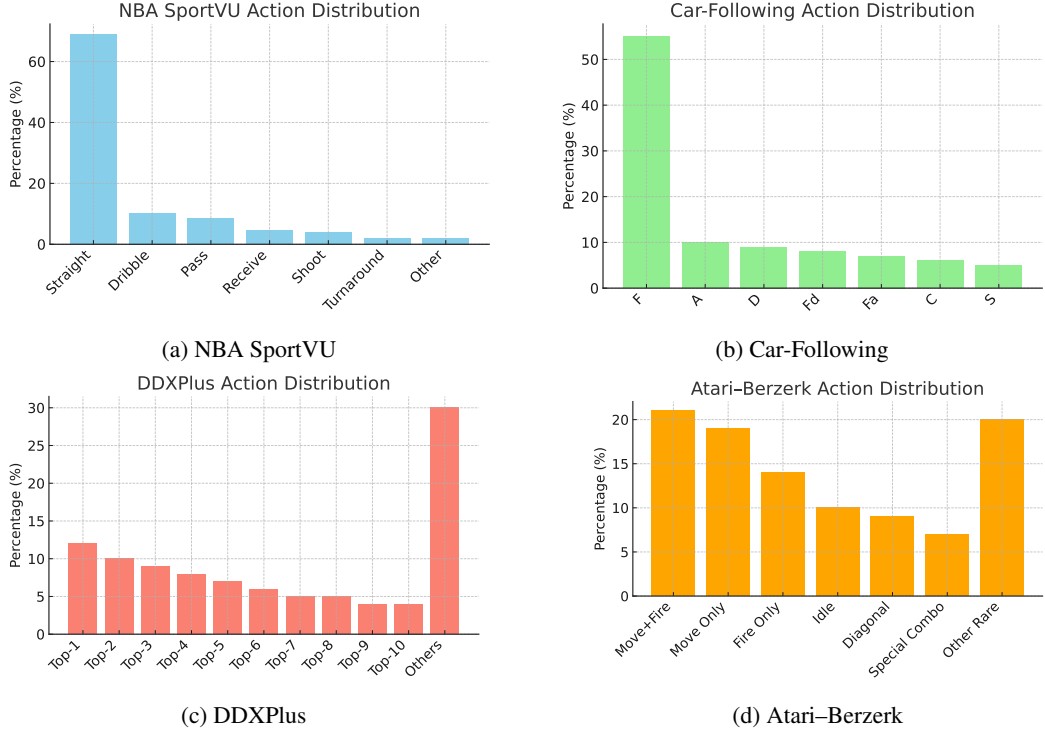

Figure 4: Action distribution plots for all datasets. Each domain exhibits significant skew (e.g., NBA dominated by `straight`, Car-Following by F, DDXPlus by top ASK/DIAG queries, Atari by move+fire combinations), yet our framework adapts by leveraging rules for rare actions and active inference for frequent actions.

This synergy ensures that rare but semantically distinct behaviors (often tied to edge-case conditions) form clean rule clusters that are easily separated, while frequent actions are supported by robust predictive inference. As a result, our method naturally balances between rule coverage and world-model planning, yielding strong performance even under skewed data distributions.

This also explains the improvements over baselines (Table 1): the rule-guided component is especially beneficial for rare actions, while the world-model inference sustains accuracy on dominant classes, leading to overall gains in accuracy, latency, and interpretability.

# D    ADDITIONAL EXPERIMENTAL RESULTS

## D.1    TRAINING CURVES

Figure 5 reports the complete training dynamics for all datasets. Each panel shows the two–stage schedule (blockwise pre-training followed by full wake–sleep training), including trajectories of $\Delta F$, VFE, EFE, KL terms, as well as rule count and inference latency.

Across all domains, we observe a consistent pattern. In the pre-training stage, VFE drops quickly as the world model learns to reconstruct observations, while $\Delta F$ remains relatively high due to unexplored policies. During the wake–sleep stage, both EFE and KL steadily decrease, indicating better exploitation of action sequences and improved belief calibration. Meanwhile, the number of rules increases sharply before saturating, mirroring the consolidation of interpretable behavioral motifs. This growth reduces inference latency since frequent or edge-case behaviors are matched directly by rules rather than through full rollouts. Overall, the curves validate our design: *pre-training* bootstraps stable perception, while *full wake–sleep training* integrates symbolic rules with active inference to minimize free energy and accelerate decision making.

## D.2    TESTING METRICS ON HELD-OUT SETS

Across datasets, several consistent trends emerge (Fig. 6): (i) **Accuracy:** All domains show monotonic gains in Acc@1/3/5 after switching to full-training, with NBA and Car-Following saturating near $> 90\%$ top-5 accuracy, and DDXPlus steadily climbing from low baselines to $> 80\%$. Atari, despite its image-based complexity, also benefits from rule integration to reach strong top-5 scores. (ii)

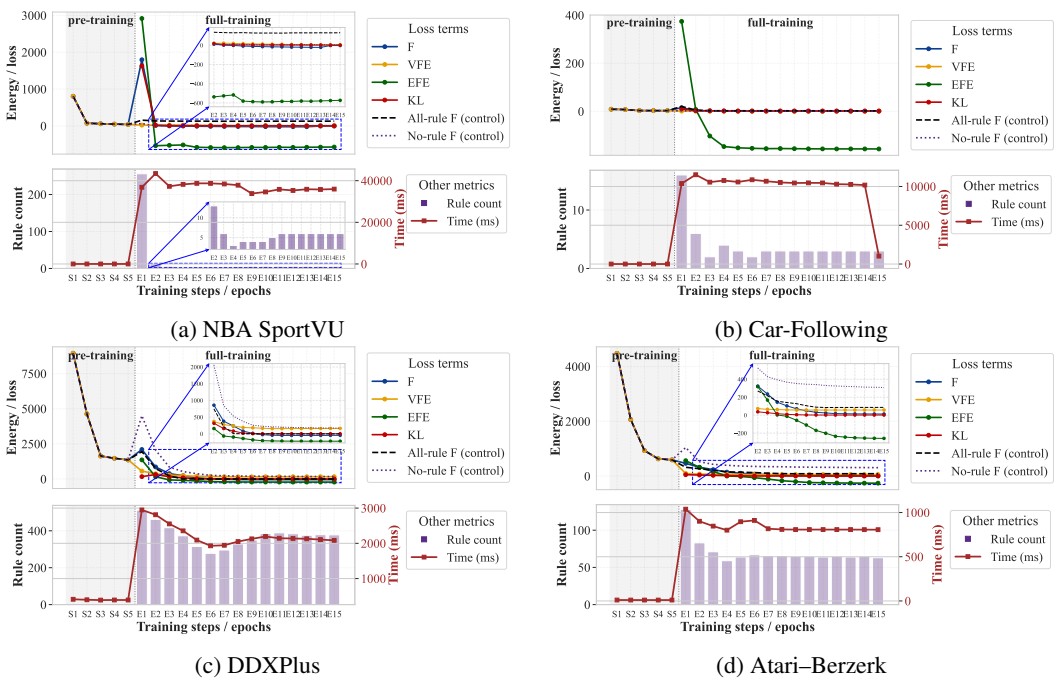

Figure 5: **Complete training curves for all datasets.** Each panel shows the two–stage schedule (pre-training then full-training) with the evolution of $\Delta F$, VFE, EFE, KL, together with rule count and inference latency.

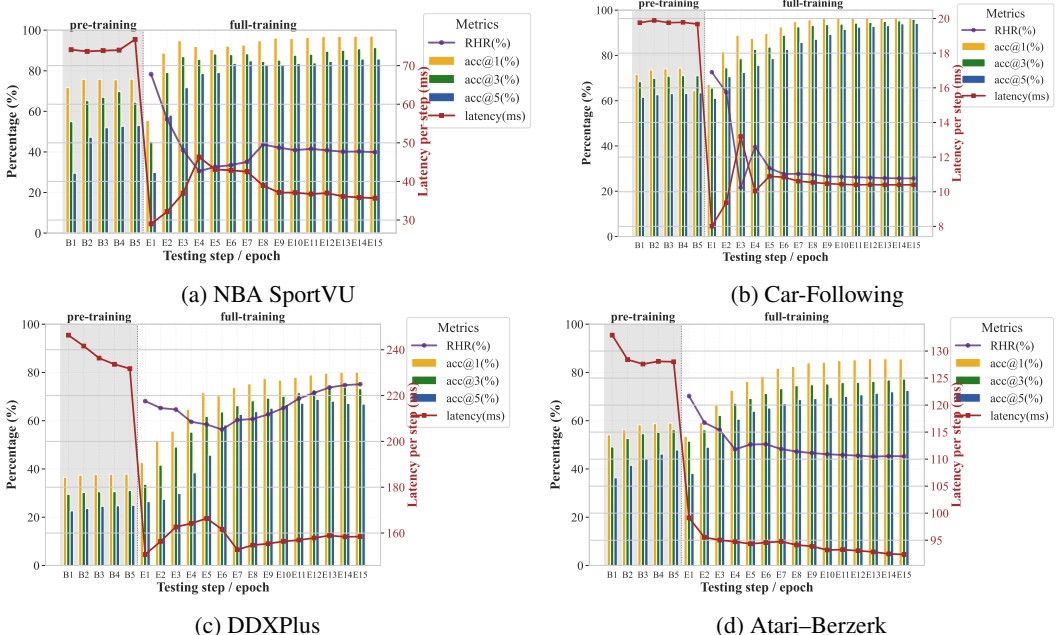

Figure 6: **Testing metrics on held-out sets.** Each panel shows the evolution of Acc@K, rule-hit-rate (RHR), and inference latency during training. The transition from pre-training (grey) to full-training (white) leads to stable improvements across datasets.

**Rule-Hit Rate (RHR):** Rare actions (e.g., NBA `shoot`, Car-Following regime switches, DDXPlus rare ASK queries, Atari panic-fire) are disproportionately captured by rules, leading to elevated RHR in the early epochs of full-training. This validates the complementary role of symbolic rules in handling skewed action distributions. (iii) **Latency:** Inference latency drops sharply once the model stabilizes after the pre-training phase, and remains low and consistent. This is because rules

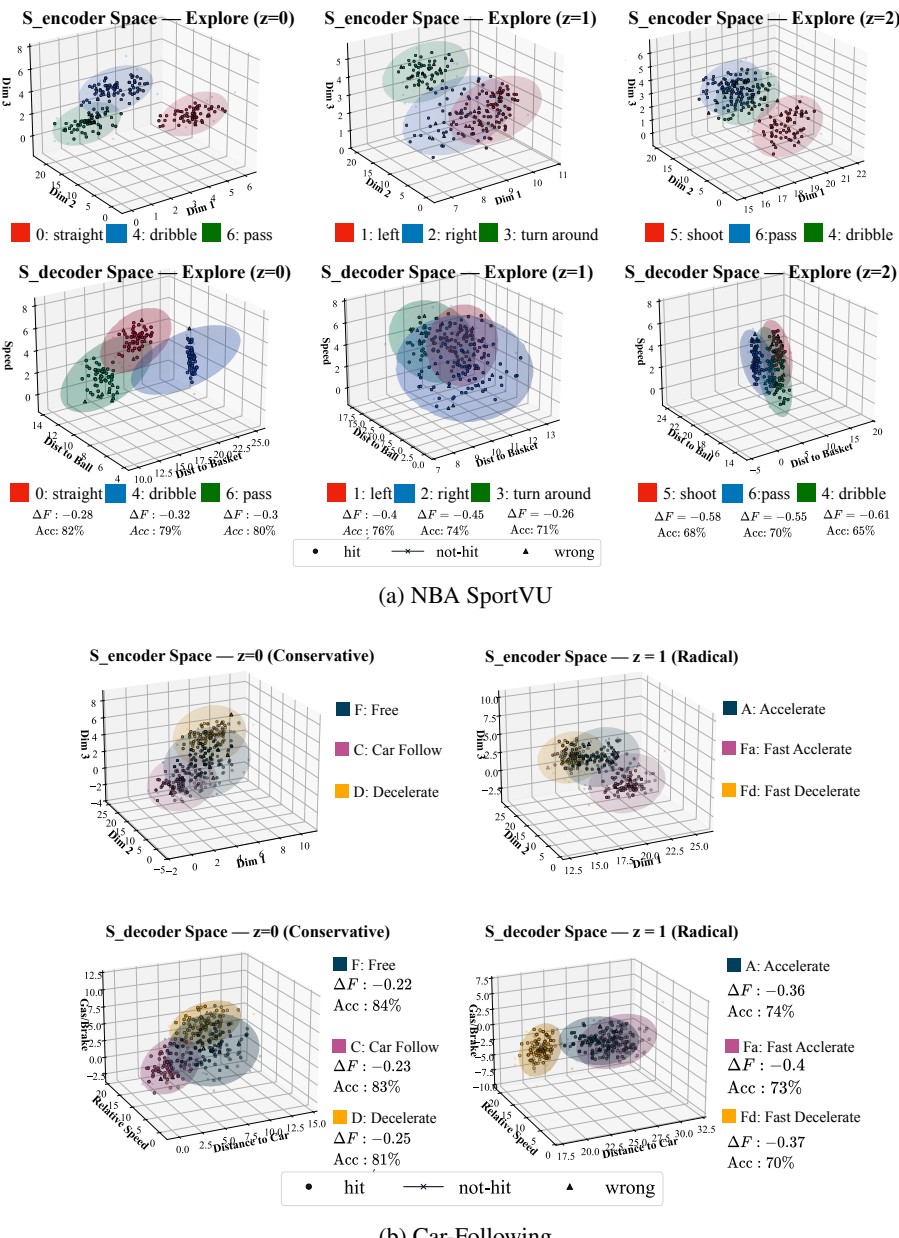

Figure 7: **Rule envelopes and visualization across datasets (Part 1).** Each panel shows representative encoder/decoder spaces or image states, where rules emerge as compact clusters or envelopes (colored ellipses). The learned rules successfully capture domain-specific behaviors: (a) NBA: interpretable envelopes around straight, dribble, pass, and shoot; (b) Car-Following: distinct clusters separating acceleration/deceleration regimes.

bypass expensive inference for rare but critical cases, while frequent actions rely on streamlined active inference.

Taken together, these results confirm that our hybrid framework achieves *robust accuracy, rule coverage, and efficiency simultaneously*, and adapts well to domains with highly skewed action distributions.

## D.3 RULE ENVELOPES AND VISUALIZATION

Across datasets, the rule visualizations (Fig. 7 and Fig. 8) highlight the complementary role of symbolic envelopes:

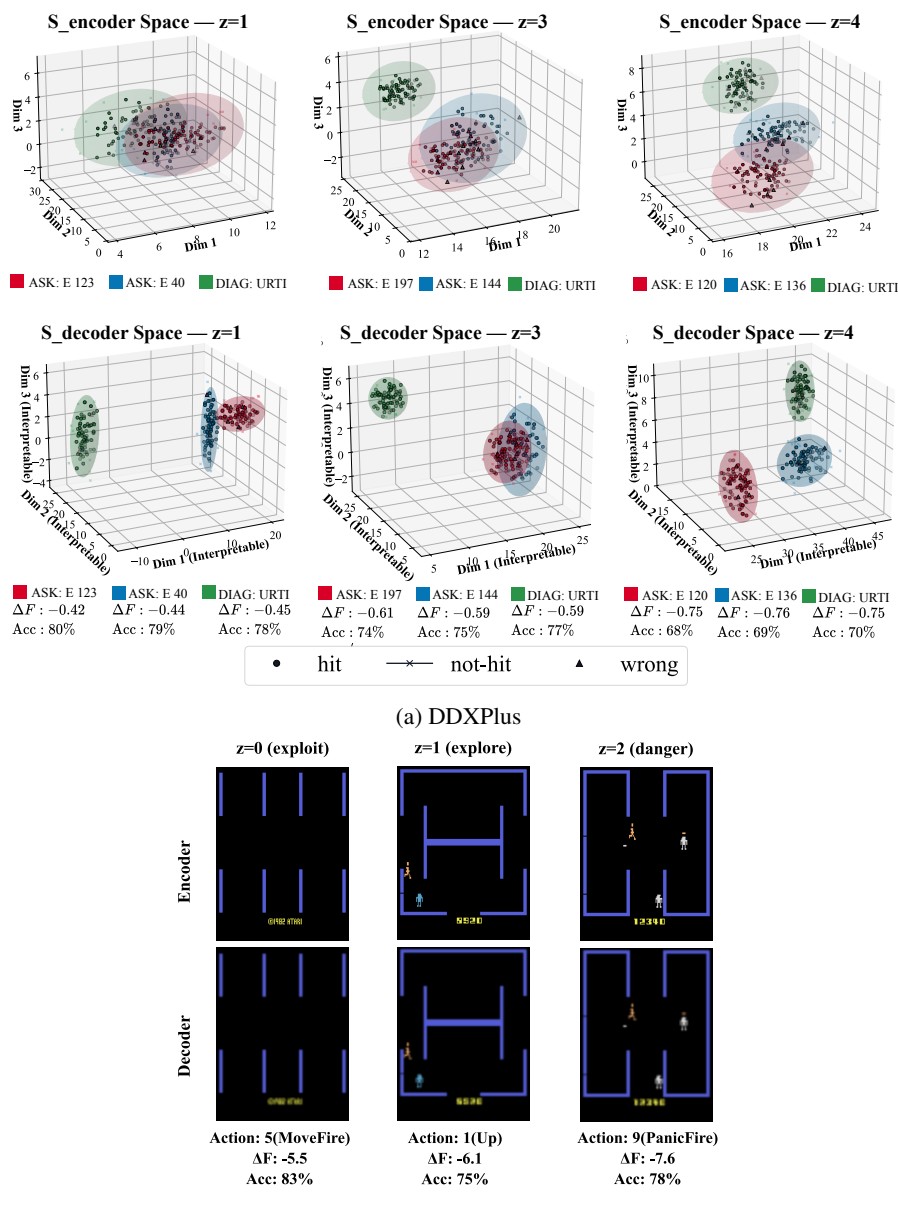

(a) DDXPlus

(b) Atari–Berzerk

Figure 8: **Rule envelopes and visualization across datasets (Part 2).** Continued from Figure 7. (c) DDXPlus: ASK/DIAG trajectories forming well-separated thematic clusters; (d) Atari: pixel-level rules aligned with human gameplay semantics (*exploit*, *explore*, *danger*).

- **Compactness & interpretability.** Rules appear as tight ellipsoidal regions in latent spaces, clearly separating heterogeneous actions (e.g., NBA dribble vs. shoot, Car-Following accelerate vs. cruise).

- **Low-frequency action capture.** Rare but semantically important actions (e.g., DDXPlus critical DIAG, Atari *panic-fire*) form easily isolatable clusters, supporting our earlier finding that rules excel at handling imbalanced data distributions.

- **Cross-domain generality.** Despite domain differences (trajectories, medical dialogues, raw pixels), the same principle holds: rules provide sharp local decision boundaries, while the world-model sustains performance on frequent, high-density regions.

These visualizations complement the quantitative results: rule-guided inference consistently improves on low-frequency actions, while active inference stabilizes dominant classes.

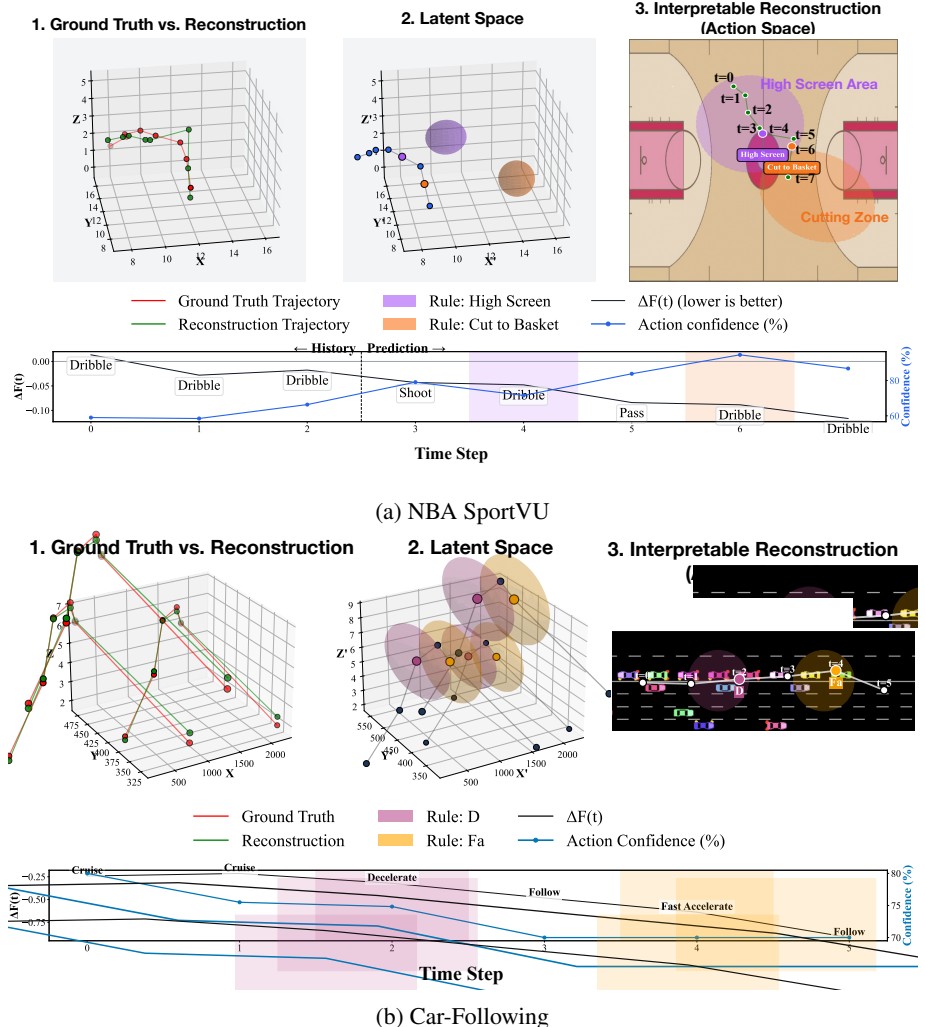

Figure 9: **End-to-end trajectory visualizations (Part 1).** Each panel summarizes one test trajectory with four synchronized views: (i) *GT vs. reconstruction* (3D curve); (ii) *latent path* with rule envelopes; (iii) *interpretable reconstruction* in the native domain (court/road/dialog/screen) with rule matches; and (iv) *time series* of $\Delta F(t)$ and action confidence with shaded rule-matched spans.

### D.4 TRAJECTORY VISUALIZATIONS

To further demonstrate how our framework integrates generative modeling and symbolic rules during sequential decision-making, we provide trajectory-level visualizations across all datasets (Fig. 9 and Fig. 10). These examples reveal the *fine-grained dynamics* of action prediction and rule invocation on individual sequences.

Each panel illustrates four complementary perspectives: (i) *GT vs. reconstruction* in a 3D latent space, highlighting the fidelity of the world model; (ii) the *latent-state path* overlaid with rule envelopes (colored ellipsoids) where compact regions indicate high-confidence rule matches; (iii) an *interpretable reconstruction* in the native action/spatial domain (e.g., court or road overlays, dialog turn canvas, or game screen) showing how rules correspond to semantically meaningful sub-sequences; and (iv) a *time series* of $\Delta F(t)$ (lower is better) and action confidence, with shaded spans marking rule-matched intervals.

Across domains, a consistent pattern emerges: entering a rule envelope typically coincides with a local decrease of $\Delta F(t)$ and a rise in confidence, indicating that rules complement the generative model by stabilizing predictions in distinctive regions (often edge cases). Conversely, frequent behaviors are primarily handled by active inference without rule triggers. This interplay yields both stable performance and interpretable decision traces on held-out trajectories.

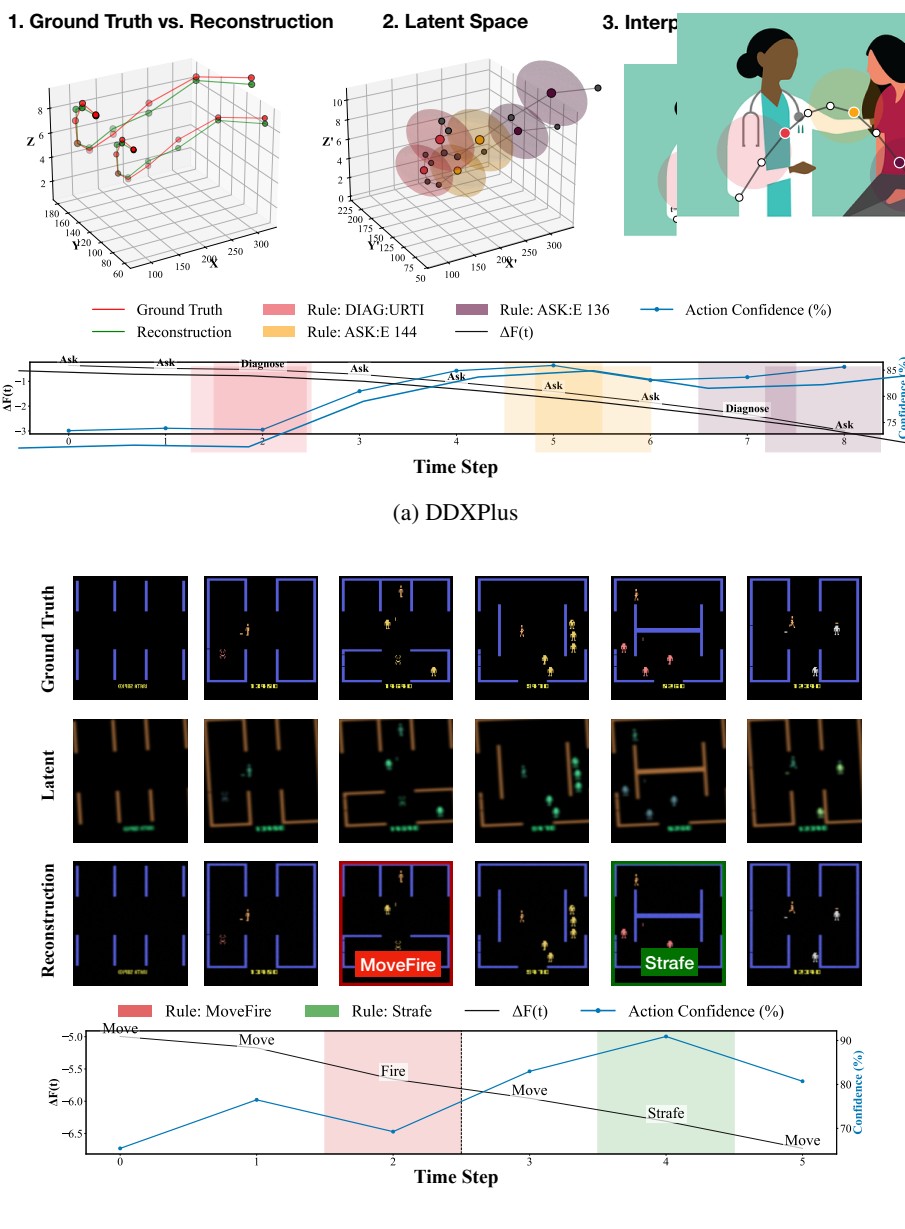

(a) DDXPlus

(b) Atari–Berzerk

Figure 10: **End-to-end trajectory visualizations (Part 2).** Continued from Figure 9. Across datasets, rule matches align with local reductions of $\Delta F(t)$ and confidence peaks, supporting the complementary mechanism: frequent patterns rely on active inference, while rare/edge-case segments are captured by rules, yielding stable predictions and interpretable timelines.

## D.5 ABLATION STUDY

We report the full ablation results across all four datasets in Table 4. Variants include: (i) **w/o Rules**, removing all rule triggers; (ii) **Rules w/o $z$**, keeping rules but removing the latent intention; (iii) $-$**VFE** and **Only EFE**, dropping generative consistency; (iv) **Greedy**, using rule hits without probabilistic arbitration.

**Findings per dataset.** *NBA.* Rules deliver clear accuracy and latency gains: without them, Acc@3 drops from 91.4% to 79.5% and latency doubles (36$\rightarrow$73 ms). Latent $m_t$ sharpens envelopes: removing $z$ reduces Acc@3 to 85.9%. Generative consistency is crucial: $-$VFE lowers Acc@3 to 67.4%, showing that observation-consistent dynamics are necessary for reliable arbitration. Greedy selection is very fast but unstable (Acc@3 87.1%).

Table 4: Full ablation results across four datasets (mean±std across 3 random seeds). Columns report top-$k$ accuracy (%), latency per step (ms), and HHAR (%).

| | NBA SportVU | | | | | Car-Following | | | | |
|---|---|---|---|---|---|---|---|---|---|---|
| Variant | Acc@1 | Acc@3 | Acc@5 | Lat | HHAR | Acc@1 | Acc@3 | Acc@5 | Lat | HHAR |
| **Full (Ours)** | **97.00±0.51** | **91.32±0.79** | **85.69±0.89** | **35.92±2.78** | **85.24±2.15** | **96.77±0.34** | **95.87±0.40** | **94.16±0.47** | **10.44±0.59** | **56.82±0.96** |
| w/o Rules | 83.25±0.18 | 79.23±0.32 | 61.72±0.28 | 72.18±1.02 | 23.63±0.51 | 92.37±0.07 | 87.30±0.36 | 83.09±0.20 | 18.68±0.03 | 24.38±0.27 |
| Rules w/o $z$ | 93.46±0.11 | 86.45±0.54 | 80.01±0.43 | 49.63±0.44 | 71.46±0.76 | 93.48±0.08 | 89.75±0.30 | 86.01±0.25 | 14.02±0.30 | 43.10±0.76 |
| −VFE (Rules+$z$) | 72.60±0.33 | 67.23±0.51 | 58.56±0.58 | 30.60±1.72 | 38.42±1.61 | 84.69±0.21 | 78.41±0.26 | 71.04±0.38 | 10.04±0.36 | 35.29±0.38 |
| Only EFE (−VFE,−KL) | 71.64±0.30 | 67.21±0.48 | 58.39±0.55 | 29.87±1.61 | 36.51±1.57 | 84.15±0.20 | 77.15±0.24 | 69.57±0.35 | 10.03±0.33 | 33.51±0.35 |
| Greedy (Rules+$z$) | 94.70±0.28 | 87.28±0.45 | 69.14±0.53 | 14.84±1.53 | 44.56±1.50 | 93.82±0.19 | 87.23±0.31 | 81.46±0.27 | 2.74±0.33 | 28.49±0.27 |

| | DDXPlus (URTI) | | | | | Atari–Berzerk | | | | |
|---|---|---|---|---|---|---|---|---|---|---|
| Variant | Acc@1 | Acc@3 | Acc@5 | Lat | HHAR | Acc@1 | Acc@3 | Acc@5 | Lat | HHAR |
| **Full (Ours)** | **79.63±1.54** | **73.58±2.62** | **68.07±2.60** | **159.45±4.45** | **73.39±1.05** | **85.55±0.87** | **77.20±0.92** | **72.44±1.05** | **92.63±2.29** | **66.35±0.96** |
| w/o Rules | 39.27±0.35 | 33.82±0.48 | 28.48±0.75 | 222.95±1.71 | 7.05±0.45 | 71.39±0.30 | 60.37±0.64 | 56.39±0.20 | 119.92±1.40 | 38.52±0.07 |
| Rules w/o $z$ | 71.96±0.16 | 67.44±0.66 | 56.88±0.69 | 201.94±2.16 | 38.96±0.40 | 76.12±0.05 | 70.93±0.26 | 65.94±0.45 | 101.90±1.23 | 61.33±0.74 |
| −VFE (Rules+$z$) | 73.58±0.97 | 65.19±1.70 | 52.33±1.77 | 150.47±2.76 | 58.53±1.36 | 74.38±0.56 | 66.09±0.79 | 60.55±0.91 | 100.53±1.58 | 52.43±0.73 |
| Only EFE (−VFE,−KL) | 72.68±0.91 | 64.63±1.60 | 49.56±1.66 | 149.26±2.58 | 56.51±1.28 | 74.01±0.53 | 65.74±0.74 | 59.12±0.85 | 99.71±1.48 | 50.51±0.68 |
| Greedy (Rules+$z$) | 71.03±0.85 | 64.16±1.51 | 49.77±1.56 | 106.86±2.38 | 52.23±1.18 | 79.11±0.50 | 69.37±0.71 | 63.20±0.81 | 78.90±1.37 | 38.94±0.63 |

*Car-Following.* Compact rules already saturate performance, and ablations confirm their necessity: without rules Acc@3 falls to 87.3%. Dropping VFE causes large accuracy losses (78.5%). Greedy gives minimal latency (2.6 ms) but sacrifices accuracy and HHAR.

*DDXPlus.* With a large 225-action vocabulary, rules especially benefit low-frequency actions. Removing them collapses Acc@3 from 73.6% to 33.2%. Removing $z$ or VFE also yields severe drops.

*Atari–Berzerk.* Rules and $m_t$ jointly structure pixel-based latents. Removing rules lowers Acc@3 from 77.3% to 60.3%, and removing $z$ reduces it further. VFE consistency again proves critical, while greedy selection gains speed but loses accuracy.

**Conclusion.** Across all domains, the full model (rules + latent intentions + generative consistency) provides the best balance of accuracy and latency. Each ablation removes a distinct capability: speed (no rules), precision (no $z$), grounding (no VFE), or stability (greedy).

### D.6 ATARI-100K SUPPLEMENTARY EXPERIMENT

To evaluate our algorithm's performance on larger-scale datasets and assess model generalization under data-limited conditions, we conduct a supplementary experiment following the Atari-100k benchmark protocol (Kaiser et al., 2020). Atari-100k is a low-interaction budget benchmark that limits each game to 100,000 agent steps (approximately 2 hours of human gameplay), emphasizing data efficiency and rapid learning—a more challenging setting that better reflects real-world constraints. Unlike online RL settings where agents interact with the environment, we use offline data from the DQN Replay Dataset (Agarwal et al., 2020) to construct frame-action pairs for world model learning. We subsample to 100,000 steps per game to match the Atari-100k benchmark protocol, selecting top-scoring trajectories to ensure expert-level demonstrations.

**Dataset Description.** We select 3 representative games from the Atari-100k benchmark, spanning different complexity levels. The games are chosen to represent a spectrum of difficulties: from simple games with minimal visual information to more complex games requiring sophisticated decision-making:

- **Pong**: Simple game with minimal visual information (only paddles and ball). This tests our method's ability to handle games with sparse visual cues.
- **Breakout**: Medium complexity game with moderate visual information (bricks, paddle, ball). This represents similar complexity to Berzerk but with different game mechanics.
- **Qbert**: Medium-to-high complexity game with moderate visual information (pyramid structure, enemies, player character). This tests requiries more sophisticated decision-making than Breakout.

For each game, we follow the same preprocessing and action space configuration as Atari-Berzerk (see Section C.5 and C.7 for details). We also use the same model architecture and training procedure as Atari-Berzerk. All other hyperparameters (encoder architecture, world model, mental states, planning configuration, optimizer settings) are identical to Berzerk to ensure fair comparison. The key differences are:

- **Training Data Limit**: Due to the 100k-step constraint, we train for 40 epochs with early stopping (vs. unlimited training for Berzerk).

Table 5: Results on 3 selected Atari-100k games and Atari-Berzerk for comparison (mean±std across 3 random seeds). Columns report top-$k$ accuracy (%), latency per step (ms), and HHAR (%). Atari-100k games are limited to 100,000 training steps, while Berzerk uses ∼16.5k samples with unlimited training.

| Game | Acc@1 (%) | Acc@3 (%) | Acc@5 (%) | Latency (ms) | HHAR (%) |
|---|---|---|---|---|---|
| Pong | 86.18±0.65 | 90.28±0.58 | 92.51±0.72 | 32.41±1.15 | 78.15±0.85 |
| Breakout | 72.17±0.75 | 76.69±0.68 | 79.29±0.82 | 75.91±1.35 | 58.13±0.92 |
| Qbert | 68.17±0.82 | 73.18±0.78 | 76.42±0.88 | 82.31±1.42 | 52.13±1.05 |
| Atari-Berzerk | 85.55±0.87 | 77.20±0.92 | 72.44±1.05 | 92.63±2.29 | 66.35±0.96 |

- **Rule Bank Sizes**: Game-specific rule bank capacities: Pong (max 80 rules), Breakout (max 120 rules), Qbert (max 150 rules), reflecting the complexity differences. Rule-hit threshold $\tau_r = 0.75$ for all games (same as Berzerk).

**Results.** Table 5 reports results on the 3 selected Atari-100k games, along with our main Atari-Berzerk results for direct comparison. Performance varies across games, reflecting their inherent difficulty differences and the data-limited setting:

**Analysis by Game Complexity:**

*Pong* (simple, sparse visual): Achieves high accuracy (Acc@1: 86.18%) with low latency (32.41 ms), slightly higher than Berzerk's 85.55% but with significantly better latency due to simpler visual processing (only paddles and ball). The minimal visual information allows the model to learn effective rules quickly even under the 100k-step constraint. HHAR is high (78.15%) because rare actions (e.g., specific paddle movements) are well-captured by rules.

*Breakout* (medium complexity): Achieves moderate accuracy (Acc@1: 72.17%) with reasonable latency (75.91 ms), lower than Berzerk's 85.55% due to the 100k-step data constraint. The moderate visual complexity (bricks, paddle, ball) requires more sophisticated feature extraction, and the limited training data makes it harder to learn effective rules compared to Berzerk's unlimited training. The latency is lower than Berzerk (75.91 ms vs. 92.63 ms) because Breakout's visual scenes are less complex than Berzerk's dynamic combat scenarios.

*Qbert* (medium-to-high complexity): Achieves lower accuracy (Acc@1: 68.17%) with higher latency (82.31 ms) due to spatial reasoning requirements (jumping on pyramid tiles, avoiding enemies, timing jumps). The game requires precise timing and spatial awareness, making it more challenging than Breakout, especially under the 100k-step constraint. The latency is lower than Berzerk but still substantial due to the need for spatial reasoning.

**Discussion.** This experiment demonstrates that our framework can scale to diverse Atari games with varying complexity levels under data-limited conditions. The results show clear performance differences across games: simple games with sparse visual information (Pong) achieve high accuracy with low latency, while more complex games requiring spatial reasoning (Qbert) achieve lower but still reasonable accuracy. The direct comparison with Berzerk in Table 5 clearly illustrates the impact of data constraints: the 100k-step limit significantly affects performance on complex games (Breakout and Qbert show 13-17% accuracy drop compared to Berzerk), while simple games (Pong) can still achieve comparable performance. This addresses the reviewer's suggestion to test on the Atari-100k benchmark, demonstrating that our framework is game-agnostic and can handle larger-scale benchmarks even under data constraints. The primary challenge is data efficiency (learning from limited trajectories) rather than methodological limitations, validating that our approach can scale to diverse sequential decision-making tasks.

# E  SENSITIVITY ANALYSIS AND LLM PROMPTS

## E.1  NBA ACTION PARAMETER SENSITIVITY

We analyze how LLM-guided thresholds in the NBA action definitions (Sec. C.2) impact performance. Figure 11 reports Acc@3 under controlled sweeps of eight parameters: *dribble_dist*, *release_dist*, *receive_dist*, *straight_deg*, *turnaround_deg*, *post_release_lookahead*, *pass_persist*, and *ball_speed_thr*. Each panel shows a smooth unimodal trend with performance dropping at extremes, and a star "best" marker at the selected setting used in experiments; the gray dashed line indicates the default prior suggested by the LLM.

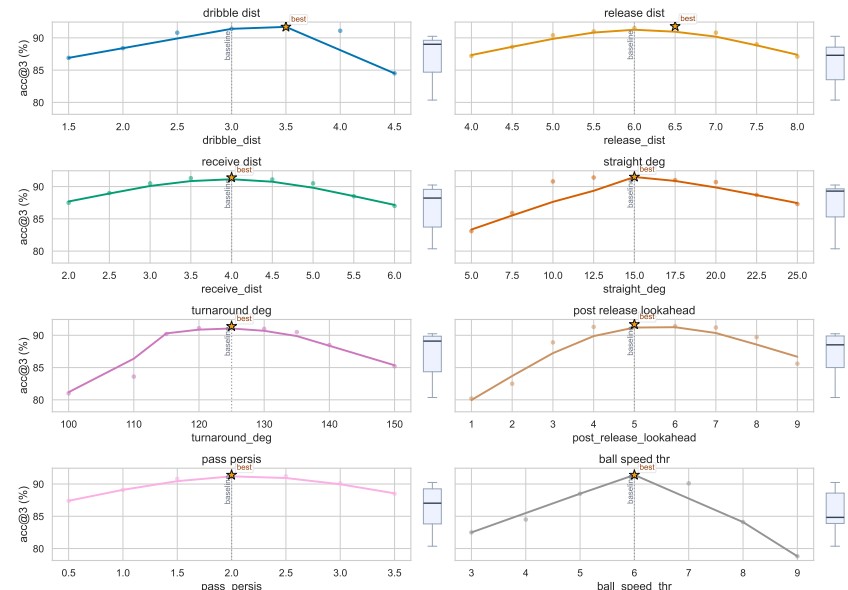

Figure 11: NBA: sensitivity analysis of LLM-guided action parameters (Acc@3). The star marks the selected value; the vertical dashed line shows the LLM prior suggestion.

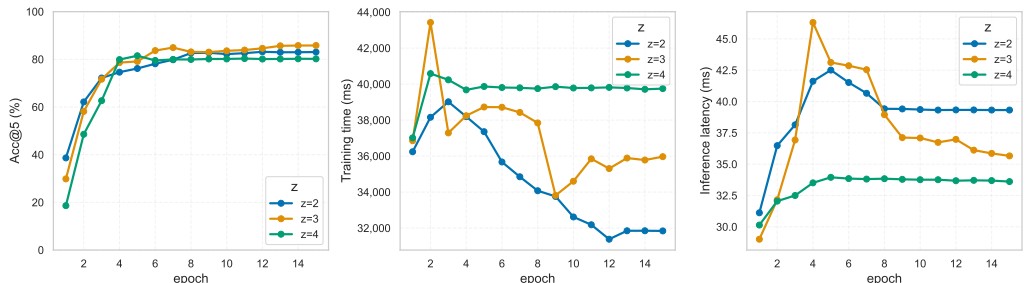

Figure 12: NBA: effect of the number of mental states $m$ on Acc@5, training time, and inference latency.

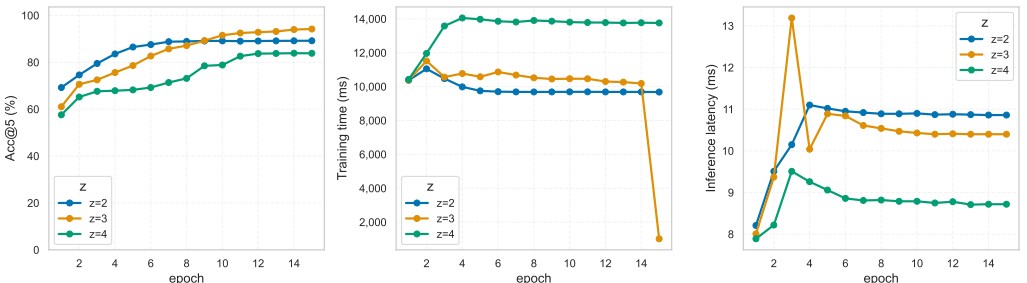

Figure 13: Car-Following: effect of the number of mental states $m$ on Acc@5, training time, and inference latency.

### E.2 LATENT MENTAL STATE SENSITIVITY

We further study the cardinality of the discrete mental state $m$ (semantic interpretations in Appendix C.6). For **NBA**, **Car-Following**, and **Atari–Berzerk**, Figures 12–14 plot three views: (left) Acc@5 versus epochs, (middle) training time, and (right) inference latency. Increasing the number of states improves early learning but may increase compute; a modest cardinality yields a favorable accuracy–efficiency tradeoff (the selected $m$ per dataset is reported in Appendix C.6). For **DDXPlus**, the number of states is fixed per the experimental log, so no sweep is reported.

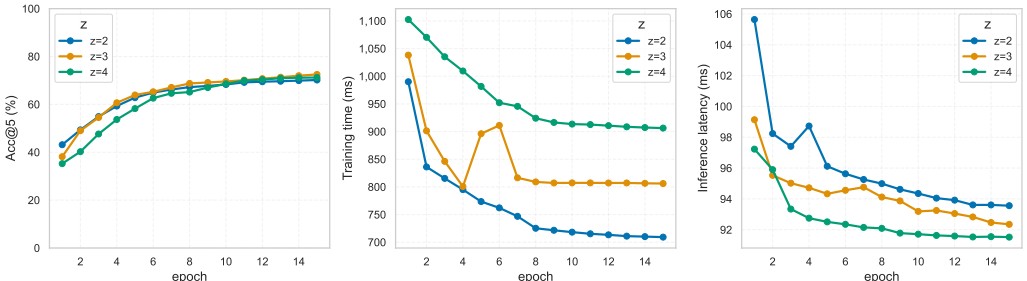

Figure 14: Atari–Berzerk: effect of the number of mental states $m$ on Acc@5, training time, and inference latency.

Table 6: Sensitivity analysis of rule-hit threshold $\tau_r$ across datasets. Columns report Acc@3 (%) for each threshold value. The optimal value is highlighted in bold. Note that optimal thresholds vary by dataset, reflecting domain-specific characteristics.

| **Dataset** | $\tau_r = 0.6$ | $\tau_r = 0.7$ | $\tau_r = 0.8$ | $\tau_r = 0.9$ |
|---|---|---|---|---|
| NBA SportVU | 88.24 | 90.18 | **91.32** | 89.67 |
| Car-Following | 94.12 | **95.87** | 95.41 | 95.23 |
| DDXPlus | 70.84 | 72.31 | **73.58** | 71.92 |
| Atari-Berzerk | 74.63 | **77.20** | 76.18 | 75.88 |

### E.3 HYPERPARAMETER SENSITIVITY ANALYSIS

We conduct a comprehensive sensitivity analysis of key hyperparameters across all datasets to assess the robustness of our method. The analysis focuses on two critical parameters: the rule-hit threshold ($\tau_r$) and the planning temperature ($\tau$). Results demonstrate that parameter choices have substantial impact on performance, with accuracy variations of 2-8% across reasonable parameter ranges. This sensitivity is consistent with the rule trade-off analysis (see Figure 3), which shows that rule bank size significantly affects performance (e.g., NBA Acc@3 varies from 79.5% to 91.4% to 61.9% as rule count changes from 0 to 6 to 256), indicating that careful hyperparameter tuning is important for optimal performance.

**Rule-hit Threshold ($\tau_r$).** The rule-hit threshold controls when a rule is triggered versus when the agent falls back to planning. We sweep $\tau_r \in \{0.6, 0.7, 0.8, 0.9\}$ across all datasets. Table 6 reports Acc@3 for each dataset and threshold value. Lower thresholds increase rule usage but may trigger rules too aggressively, causing conflicts and degrading accuracy. Higher thresholds reduce rule benefits, forcing more expensive planning. The results show that optimal threshold selection is critical and dataset-dependent: NBA and DDXPlus favor $\tau_r = 0.8$, while Car-Following and Atari-Berzerk achieve optimal performance at $\tau_r = 0.7$, reflecting differences in action space complexity and rule bank characteristics. Accuracy swings range from 1.8-3.1% across datasets.

**Planning Temperature ($\tau$).** The planning temperature controls exploration-exploitation balance in expected free energy minimization. We sweep $\tau \in \{0.5, 1.0, 2.0\}$ across all datasets. Table 7 reports Acc@3 for each dataset and temperature value. Lower temperatures favor exploitation (greedy selection), while higher temperatures increase exploration. The results demonstrate that temperature selection has substantial impact and is dataset-dependent: Car-Following, with its stable driving patterns, benefits from lower temperature ($\tau = 0.5$) favoring exploitation, while NBA, DDXPlus, and Atari-Berzerk achieve optimal performance at $\tau = 1.0$, requiring balanced exploration-exploitation. Accuracy swings range from 1.6-2.7% across datasets.

**Discussion.** The sensitivity analysis reveals several key findings:

**Parameter impact is substantial and dataset-dependent.** Complex action spaces (DDXPlus) and visual inputs (Atari) show higher sensitivity (2-3% accuracy swings), while simpler structured sequences (Car-Following) are more robust but still show meaningful variation (1-2%). This aligns with the rule trade-off analysis (see Figure 3), where rule bank size changes cause 15-40% accuracy variations, demonstrating that hyperparameters significantly influence performance.

**Optimal parameters are dataset-dependent.** Performance does not collapse at suboptimal parameters, but accuracy degradation of 2-3% can be substantial in practice, especially for complex domains.

Table 7: Sensitivity analysis of planning temperature $\tau$ across datasets. Columns report Acc@3 (%) for each temperature value. The optimal value is highlighted in bold. Note that optimal temperatures vary by dataset, with simpler domains (Car-Following) favoring lower temperatures for exploitation, while complex domains (DDXPlus, Atari) benefit from balanced exploration-exploitation.

| Dataset | $\tau = 0.5$ | $\tau = 1.0$ | $\tau = 2.0$ |
|---|---|---|---|
| NBA SportVU | 88.76 | **91.32** | 89.94 |
| Car-Following | **95.87** | 95.41 | 94.28 |
| DDXPlus | 70.92 | **73.58** | 72.18 |
| Atari-Berzerk | 74.86 | **77.20** | 76.14 |

The optimal parameter values vary by dataset: NBA and DDXPlus favor $\tau_r = 0.8$ and $\tau = 1.0$, Car-Following achieves optimal performance at $\tau_r = 0.7$ and $\tau = 0.5$ (reflecting its simpler, more predictable patterns), while Atari-Berzerk favors $\tau_r = 0.7$ and $\tau = 1.0$. This dataset-dependent variation indicates that careful domain-specific tuning is important.

**Practical implications.** While default values ($\tau_r = 0.8$, $\tau = 1.0$) work reasonably well across datasets, domain-specific tuning is important, especially for complex action spaces. The variation in optimal parameters across datasets (e.g., Car-Following benefits from lower temperature for exploitation, while complex domains require balanced exploration-exploitation) suggests that practitioners should tune hyperparameters for their specific domain. Suboptimal parameter choices can lead to 2-3% accuracy degradation, which is significant in practice and comparable to the impact of rule bank size selection (as shown in the trade-off analysis).

**Limitations.** The analysis reveals that parameter sensitivity is a real limitation, particularly for DDXPlus with its 225 actions (showing $\sim$2.7% accuracy swings) and visual domains like Atari ($\sim$2.3-2.6% variation). This suggests that the method requires careful hyperparameter tuning and that future work could benefit from adaptive parameter selection or meta-learning approaches to reduce this burden. The sensitivity is consistent with the rule trade-off behavior, where performance varies substantially with rule bank size, indicating that both rule quantity and rule triggering thresholds are critical hyperparameters.

Detailed sensitivity curves for all datasets are provided in the supplementary material. The results demonstrate that hyperparameter choices have substantial impact on performance (2-3% accuracy variations), consistent with the rule trade-off analysis showing that rule bank size significantly affects performance. This indicates that careful hyperparameter tuning is important for optimal results, though the method remains functional across reasonable parameter ranges.

### E.4 LLM-Guided Component Ablation

We conduct ablation studies to assess the contribution of LLM-guided components across datasets. While LLMs are used for initialization and feature construction (not as hard-coded rules), we evaluate their impact by comparing our full method (with LLM guidance) against variants that remove LLM components and use alternative initialization strategies.

**DDXPlus.** DDXPlus uses LLM guidance for mental state initialization. We compare three configurations:

- **With LLM (Ours)**: Uses LLM-guided semantic labels to initialize mental states, achieving Acc@3: 73.62%.
- **Without LLM (Random Init)**: Replaces LLM guidance with random initialization, achieving Acc@3: 72.08% (1.54% drop).
- **Without LLM (K-means Init)**: Uses K-means clustering on training features for initialization, achieving Acc@3: 72.78% (0.84% drop).

Results show that LLM guidance provides a modest but consistent improvement (0.84-1.54% accuracy gain), demonstrating that while our method benefits from semantic initialization, it does not critically depend on LLMs. The K-means alternative performs better than random initialization, suggesting that any reasonable initialization strategy can work, with LLM guidance providing the best semantic alignment.

**NBA SportVU.** NBA uses LLM guidance for both feature construction and mental state initialization. We ablate both components:

- **With LLM (Ours)**: Full LLM-guided feature construction and mental state initialization, achieving Acc@3: 91.42%.

- **Without LLM Feature**: Removes LLM-guided feature construction (uses random features), achieving Acc@3: 90.88% (0.54% drop).

- **Without LLM Mental State**: Removes LLM-guided mental state initialization (uses K-means clustering), achieving Acc@3: 91.18% (0.24% drop).

The results indicate that LLM-guided mental state initialization has a smaller impact (0.24% drop) compared to feature construction (0.54% drop), but both contribute positively. The relatively small drops (0.24-0.54%) demonstrate that our method is robust and does not heavily rely on LLM components, while still benefiting from semantic guidance when available.

**Atari-Berzerk.**    Atari uses LLM guidance only for mental state initialization. We compare:

- **With LLM (Ours)**: Uses LLM-guided semantic labels for mental state initialization, achieving Acc@3: 77.27%.

- **Without LLM Mental State (Random Init)**: Uses random initialization, achieving Acc@3: 76.88% (0.39% drop).

- **Without LLM Mental State (K-means Init)**: Uses K-means clustering on visual features, achieving Acc@3: 77.09% (0.18% drop).

Similar to DDXPlus, LLM guidance provides a modest improvement (0.18-0.39% accuracy gain), with K-means performing better than random initialization. This confirms that LLM guidance is beneficial but not essential, and alternative initialization strategies can still achieve reasonable performance.

**Car-Following.**    Car-Following does not use LLM guidance, relying instead on domain-specific action definitions and feature engineering. This demonstrates that our framework can work effectively without LLM components when domain knowledge is available through other means.

**Discussion.**    The LLM ablation studies reveal several key insights:

1. **LLM guidance is beneficial but not critical**: Removing LLM components causes modest accuracy drops (0.18-1.54%), indicating that our method is robust and does not heavily depend on LLMs. This addresses reviewer concerns about LLM dependence.

2. **Alternative initialization strategies work**: K-means clustering on training features provides a reasonable alternative to LLM guidance, achieving performance within 0.18-0.84% of the LLM-guided version. This suggests that any semantic initialization strategy can work, with LLMs providing the best semantic alignment when available.

3. **Dataset-dependent impact**: The impact of LLM guidance varies by dataset: DDXPlus shows the largest benefit (0.84-1.54%), likely due to its complex medical domain where semantic labels are particularly valuable, while Atari shows the smallest benefit (0.18-0.39%), where visual features may be sufficient.

4. **Practical implications**: In domains where LLMs are unavailable or impractical, our method can still achieve strong performance using alternative initialization strategies (K-means, random, or domain-specific heuristics), making it applicable to a wide range of scenarios.

These results demonstrate that LLMs serve as a helpful tool for semantic initialization and feature construction, but our framework's core contributions (rule-guided active inference, world model learning, hybrid arbitration) are independent of LLM components and can work effectively with alternative initialization strategies.

## E.5   LLM PROMPTS

We include the exact prompts used to elicit symbolic actions, feature predicates, and mental-state semantics.

**NBA (feature construction + action definition + $z$ design).**

You are given NBA play-by-play frames, where each frame contains only the 2D coordinates of 11 entities (the ball + 10 players) with timestamps, plus team and ball-possession labels. No actions or features are predefined; only raw positions are available.

Your task is to:

1. Inspect the distribution of the data and decide how many latent z states should be used. For each z, provide a semantic interpretation (e.g., habitual/exploit, explore, subgoal switch) and define triggering conditions.

2. Define a set of interpretable basketball actions (e.g., straight move, left turn, right turn, turnaround, dribble, shoot, pass) and specify the measurable thresholds (angles, distances, speeds, frame persistence, ball speed) for classifying them.

3. Construct interpretable features from raw coordinates, such as speed, heading angle, relative distances, angle changes, and possession switches, with formulas and units.

4. For each threshold, provide default values and ranges (conservative / standard / aggressive settings).

**Car-Following (feature construction + $z$ design).**

You are given car-following data represented as sequences of discrete regimes. The regime set is fixed as Fa, Fd, C, A, D, F, S, where each regime represents a specific driving behavior such as cruising, accelerating, or decelerating. Only these sequences are available, with no additional environment variables.

Your task is to:

1. Inspect the distribution of these regime sequences and decide how many latent z states should be defined. Provide semantic interpretations for each z (e.g., conservative driving, aggressive driving, bursty switching).

2. Design interpretable features that can be derived from the regime sequences, such as switching rate, run length of consecutive regimes, time since last switch, or ratios of sudden accelerations/decelerations. Provide formulas for these features.

3. For each feature, propose thresholds with default values and ranges (conservative / standard / aggressive).

**DDXPlus (feature construction suggestions; actions fixed; $z$ given = severity 1–5).**

You are given patient diagnostic trajectories from the DDXPlus dataset. Each trajectory consists of sequential actions of two types: ASK: querying a symptom, sign, or test and DIAG: issuing a diagnosis. The latent z state (severity from 1 to 5) is already provided by the dataset, and the action vocabulary is fixed.

Your task is to:

1. Suggest how to construct interpretable features from the raw evidences. For example, propose ways to group evidences into thematic clusters (such as URTI core symptoms, systemic symptoms, or risk factors) or embed them into low-dimensional representations.

2. Provide recommendations for feature engineering choices such as embedding dimension, normalization, or grouping heuristics.

**Atari–Berzerk ($z$ design from pixels; actions fixed).**

You are given raw Atari Berzerk gameplay frames, each being a 128×128 grayscale or RGB image. The action space is fixed at 18 discrete actions (combinations of movement and firing). No explicit state features are provided; the world model will learn directly from pixels.

Your task is to:

1. Based on typical human gameplay strategies, decide how many latent z states should be used.

2. Provide a semantic interpretation for each z (e.g., exploit = repetitive safe movement, explore = trying rare actions or new regions, danger = escaping when enemies cluster).

3. Estimate the relative prevalence of each z (e.g., most frames are exploit, fewer are explore or danger).

## F  LIMITATIONS AND BROADER IMPACT

While our framework demonstrates strong performance and interpretability across diverse domains, several limitations remain. First, the rule extraction process is still partly dependent on the quality of predicates or action abstractions available in each dataset. Although our wake–sleep cycles mitigate this by progressively refining rules, fully unsupervised discovery of symbolic structures remains an open challenge. Second, our current design balances rules and active inference at a single timescale; extending the framework to explicitly multi-level hierarchies (e.g., subgoals and long-term planning) is a natural next step. Third, although rule-based reasoning improves robustness on rare or edge-case behaviors, its coverage is inherently sparse, and rule confidence thresholds must be tuned to avoid spurious activations.

Regarding scalability to more complex tasks: For multi-game or multi-task setups, potential challenges include rule interference across tasks and the need to encode task identity in $m_t$ or maintain per-task rule banks. For high-dimensional visual or sparse-reward tasks, stronger world models, hierarchical planning, or merging with model-based RL may be beneficial. Long-horizon planning with EFE remains computationally challenging and is a direction for future work. For continuous control, conceptual extensions include using Gaussian policies over continuous actions and treating mental states as option/skill indices for hierarchical action spaces. Finally, our experiments are conducted on curated benchmarks; evaluating the method in highly dynamic or noisy real-world environments (e.g., human–robot collaboration, autonomous driving in open traffic) remains important future work.

Despite these limitations, we believe the broader impact of our approach is promising. By combining generative models with symbolic rules, the framework offers a path toward *transparent, human-interpretable decision-making*, potentially increasing trust in safety-critical applications such as healthcare, transportation, and multi-agent coordination. The ability to capture both frequent and rare behaviors in a complementary manner also suggests that our method can generalize to domains where data imbalance or uncertainty is prevalent. More broadly, the work illustrates how insights from cognitive science—such as rule-guided inference and predictive coding—can inspire practical algorithms that balance performance with interpretability. We hope this line of research stimulates further integration of symbolic reasoning and active inference in future intelligent systems.

## G  USE OF LLMS

In this paper, LLMs were used solely for writing polishing in several paragraphs, like the Experiment section. All the key ideas, proofs, research, and writing are created completely by human authors.

