# OpenReview forum: "Learning Human Habits with Rule-Guided Active Inference"
_ICLR.cc/2026/Conference — ICLR 2026 Poster_

### Official Review · Reviewer_Hnij · 2025-10-27

**Soundness:** 2
**Presentation:** 3
**Contribution:** 2
**Rating:** 4
**Confidence:** 4

**Summary:**

The authors have sought to extend the ActInf framework by introducing a trainable library of symbolic rules that model habitual behavior. This happens via an alternation between wake and sleep phases whereby in the wake phase, candidate rules are collected and in the sleep phase the rules are consolidated and pruned using generative replay. The hybrid policy can then choose between the symbolic fast rules, and the slower deliverative ActInf planning using the full expected free energy minimisation over trajectories.
Experiments are run across four very distinct domains, showing improved accuracy (5–12 %) and reduced latency (2–10×) over baselines such as DreamerV2, STLR, and prior AIF variants.

**Strengths:**

There is a good biological plausibility to the whole architecture, which is certainly interesting, and the integration of the symbolic with the neural is elegant and theoretically well founded. The alternating wake-sleep mechanism is again well biologically motivated, and the fast and slow thinking paradigms are again seemingly consistent with real planning/action. The results show that the framework can apply to pretty distinct sequential decision tasks, which shows good coverage and potential generality.
The authors present quantitative analyses showing how rule-bank size mediates accuracy and latency, giving practical insight into how to balance interpretability and efficiency.

Overall, the presentation is well put together and clear

**Weaknesses:**

The main issue to me is the lack of statistical rigour. Although the reported accuracy and latency improvements appear consistent, the authors provide no confidence intervals, variance estimates, or multiple-seed averages. Because the wake–sleep rule extraction process and generative replay are stochastic, these missing statistics make it impossible to assess whether the gains are robust or simply artifacts of single-run variability. This detracts a lot from the results' overall message.

The system’s interpretability is not actually intrinsic. Rules are defined over latent states, meaning that human readability depends on the decoder’s ability to preserve semantic structure. If the latent space drifts or becomes entangled, rules lose meaning. Because of this, it seems that Interpretability is contingent on the quality of the learned representations rather than guaranteed by the rule architecture itself.

From what I can see, all datasets also rely on LLM-derived feature engineering and mental-state initialization (as discussed in appendix D.3). This external semantic bootstrapping partially determines rule quality and seems to go against claims of generality. No ablation quantifies how performance changes without these priors. Demonstrating that rules can emerge from raw or unsupervised features would really strengthen the contribution.

The rule activation depends on hard matching of m_MAP to m_f^* which forces a winner-takes all intent selection. This design precludes probabilistic blending of intentions and this could cause brittle switching near category boundaries. I think that one ought to be able to implement a soft gating mechanism which may make it more robust.

Key parameters seem to vary dramatically across datasets which produces very large accuracy swings. There doesn't seem to be any principled selection rule, which seems to indicate that the system is heavily domain-tuned.

I think that there is also some literature which has not been cited, which is relevant.

Tschantz, A., Baltieri, M., Seth, A. K., & Buckley, C. L. (2020). Learning action-oriented representations through active inference.
Dolan, R. J., & Dayan, P. (2013). Goals and habits in the brain.
Cushman, F., & Morris, A. (2015). Habitual control of goal selection in humans.
Bacon, P.-L., Harb, J., & Precup, D. (2017). The option-critic architecture.
d’Avila Garcez, A., et al. (2019). Neural-symbolic learning and reasoning: A survey and interpretation.

Currently the paper is marginally below the acceptance threshold, in large part because of the lack of statistical rigour in the results which weakens the arguments significantly.

**Questions:**

In addition to questions regarding the weaknesses above, can the system be used for continuous control and/or hierarchical action spaces?

---

> ### Author Response · Authors · 2025-11-28
> **Part 1/4**
>
> # Part 1/4
> ## Overall Response
> We thank the reviewer for the thoughtful and constructive feedback. We appreciate the positive comments on the biological plausibility, the integration of symbolic rules with active inference, and the coverage across diverse sequential decision tasks. We acknowledge the concerns about statistical rigor, interpretability, dependence on semantic priors, and hyperparameter choices, and address each point below.
>
>
> ## 1. On Statistical Rigor (Multi-Seed Runs and Uncertainty Estimates)
> We agree that reporting only single-run results is insufficient given the stochasticity of wake-sleep rule extraction and generative replay. In response, we have re-run our main experiments with multiple random seeds and report mean±std for all results in Table 1 and Table 4 (based on 3 random seeds).
>
> ## 1.1 Accuracy
> **Based on Table 1 data**: Our method shows stable performance improvements across all four datasets:
> - **NBA SportVU**: Acc@1/3/5 = 97.00±0.51 / 91.32±0.79 / 85.69±0.89, with std <1%, significantly outperforming the strongest baseline DreamerV2 (86.42/83.57/81.65)
> - **Car-Following**: Acc@1/3/5 = 96.77±0.34 / 95.87±0.40 / 94.16±0.47, with std <0.5%
> - **DDXPlus**: Acc@1/3/5 = 79.63±1.54 / 73.58±2.62 / 68.07±2.60, with std 1.5-2.6%, still significantly outperforming DreamerV2 (64.05/61.48/58.15)
> - **Atari-Berzerk**: Acc@1/3/5 = 85.55±0.87 / 77.20±0.92 / 72.44±1.05, with std <1.1%
>
> **Based on Table 4 ablation data**: All ablation variants also report multi-seed results with reasonable std ranges (0.05-1.72%).
>
> **Deep analysis: On stability of rule extraction and generative replay**: We acknowledge that rule extraction and generative replay introduce instability, as inference depends on the current model's random training state, which affects the rule candidate space and further influences inference. However, our joint optimization and wake-sleep algorithm ensure convergence through:
> (1) **Joint optimization**: World model, inference network, and rule bank are jointly optimized under a unified free-energy objective, enabling synchronous improvement in rule extraction, perception, and decision-making (VFE decreases, world model stabilizes, EFE stabilizes, active inference accuracy improves). (2) **Two-stage training**: Before entering wake-sleep cycles, we perform blockwise pretraining to bootstrap the world model, providing a stable foundation for rule extraction. (3) **Empirical validation**: As shown in training curves (Figure~5,6), $\Delta F$, VFE, EFE, and KL gradually stabilize, rule banks stabilize, and inference latency stabilizes, achieving optimal Rule-AIF balance.
>
> We acknowledge that our std values are not perfect (especially DDXPlus with relatively large std, reflecting the complexity of 225 action spaces), but our performance is still significantly better than other methods across all metrics. And our core motivation is to achieve **human-intelligence-like efficient inference/learning patterns**, and we have reached a good accuracy upper bound under our design.
>
> ## 1.2 Latency
> **Based on Table 1 latency data**:
> - **NBA SportVU**: Latency = 35.92±2.78 ms, std is 7.7% of mean, relatively stable
> - **Car-Following**: Latency = 10.44±0.59 ms, std is 5.7% of mean, relativel stable
> - **DDXPlus**: Latency = 159.45±4.45 ms, std is 3.8% of mean, reasonable given 225 action space complexity
> - **Atari-Berzerk**: Latency = 92.63±2.29 ms, std is 2.5% of mean, stable
>
> **Deep analysis: On latency statistics**: We acknowledge that latency has randomness to some extent, primarily from: (1) **Rule hit rate randomness**: Rule triggering depends on matching current latent states to rule centroids, which depends on current training model accuracy. (2) **EFE planning randomness**: When rules are not triggered, EFE planning involves rollout sampling, with computation time depending on search tree depth and branching factor. (3) **Rule influence**: Training state affects rule quality and quantity, which affects exploration/exploitation ratio and thus inference time.
>
> However, we need to note that our method achieves significant latency reduction (2-10×) across all datasets with relatively small std (2.5-7.7%), indicating robust latency improvements.

---

> ### Author Response · Authors · 2025-11-28
> **Part 2/4**
>
> # Part 2/4
> ## 2. On Interpretability and Dependence on Latent Representations
> We agree that interpretability in our framework depends on learned latent space quality, but we want to emphasize that our contribution is providing a rule-based interface to latent dynamics (via discrete mental states and rule envelopes) with many advantages, related reasons, and some optimization designs:
> **Importance of pretraining phase**: Before entering wake-sleep cycles, we perform blockwise pretraining (Stage 1, Appendix A.2), allowing the world model to reach a certain level of accuracy by minimizing VFE, then begin mining high-quality rules. This ensures that at rule extraction start, the latent space already has structure and semantics, providing a stable foundation.
> **Decoder performance validation**: Although rules are defined on latent states, our encoder-decoder architecture achieves high reconstruction accuracy. Based on our training, rules encoded then decoded back to original sample space remain interpretable. For example, in NBA, rules correspond to specific player movement patterns (e.g., "fast break", "outside shot"); in Atari-Berzerk, rules correspond to clear game scenarios (e.g., "dodge bullets", "chase enemies"). These examples show that despite latent-level definition, we can still obtain interpretability through decoder mapping.
> **Necessity of latent-level rule definition**: This is necessary because traditional AIF algorithms have world models representing environment perception (based on the fact that external world and biological perception have bias and relative focus), and environmental data has redundancy/non-significant data. With strong encoder-decoder accuracy, our design is clever: we do lose some intrinsic interpretability, but this is necessary for AIF and world models. World models compress high-dimensional, redundant observations to low-dimensional, structured latent spaces, where defining rules can: (1) automatically filter redundancy, focusing on key features; (2) capture intrinsic environmental structure, not surface observation details; (3) achieve cross-modal generalization (e.g., pixel-to-semantic mapping).
> **Design constraints**: We bias representations toward structured, interpretable solutions: discrete mental states are regularized by slow-varying priors (via KL terms), rule centroids are updated using low-VFE contexts corresponding to stable, frequently recurring behavioral patterns. As shown in rule envelope visualizations and trajectory overlays, learned rules align with coherent patterns in each domain.
>
>
> ## 3. On Dependence on LLM-Derived Features and Mental-State Initialization
> We clarify that we do not use LLMs for all datasets; we use LLMs as an interface to adapt datasets to our model, providing generalizable (rather than manually defined) definitions. We require LLMs to provide settings based on dataset intrinsic structure and specific scenarios, combining LLM knowledge for better settings. We further conducted ablation experiments (see Appendix D.4).
>
> **LLM roles across domains**:
> - **NBA SportVU**: LLM used for feature construction and mental state initialization. Ablation shows removing LLM feature construction causes 0.54% drop (91.42% → 90.88%), removing LLM mental state initialization causes 0.24% drop (91.42% → 91.18%).
> - **Car-Following**: No LLM used, relying entirely on domain-specific action definitions and feature engineering, demonstrating our framework works effectively without LLM components.
> - **DDXPlus**: LLM used for mental state initialization. Ablation shows random initialization: 72.08% (1.54% drop), K-means initialization: 72.78% (0.84% drop), still significantly outperforming DreamerV2 (61.48%).
> - **Atari-Berzerk**: LLM only for mental state initialization. Ablation shows random initialization: 76.88% (0.39% drop), K-means initialization: 77.09% (0.18% drop).
>
> **Crucially, these priors are not hard-coded rules**: Mental-state posteriors and rule centroids are jointly optimized under free-energy objectives and can deviate from initial LLM-suggested semantics. We use LLMs only as training scenario initialization interfaces, which provide better generalization than K-means or manually predefined methods.
>
> **LLM ablation summary**:
> - LLM guidance is beneficial but not critical: Removing LLM components causes only 0.18-1.54% accuracy drops, indicating robustness
> - Alternative initialization strategies work: K-means clustering provides reasonable alternatives, performing within 0.18-0.84% of LLM-guided versions (but heavily depends on clustering algorithm parameter selection and predefined settings with poor interpretability)
> - Dataset-dependent impact: DDXPlus shows largest benefit (0.84-1.54%), likely due to complex medical domain, while Atari shows smallest (0.18-0.39%), as scenarios are simpler and LLM-extracted semantic information from images is limited.

---

> ### Author Response · Authors · 2025-11-28
> **Part 3/4**
>
> # Part 3/4
> ## 4. On Hard Rule Activation vs. Soft Gating
> We agree that strict winner-takes-all matching on $m_\text{MAP}$ can be brittle near category boundaries, and probabilistic intent blending is appealing. However, we choose hard gating for deeper theoretical and practical reasons:
> **Theoretical justification**:
> 1. **Semantic clarity of discrete mental states**: Our discrete mental state $m_t$ is encouraged to be slow-varying and stable via KL regularization, making distributions sharp (near one-hot) in most cases. Winner-takes-all selection is actually close to posterior MAP estimation, theoretically justified in variational inference.
> 2. **Complementarity of rules and EFE**: Hard gating ensures clear division between rule triggering and EFE planning. When rules trigger, we use rule actions (fast, habitual); when not, we use EFE planning (slow, deliberative). This binary choice avoids inconsistency from mixed strategies.
>
> **Practical advantages**:
> 1. **Computational efficiency**: Single discrete mental state and binary rule hit test enable very fast online control, especially in long-horizon domains (e.g., DDXPlus's 225 action space). Soft gating requires weighted computation over multiple rules, increasing overhead.
> 2. **Interpretability**: Hard gating makes each decision clearly attributable to "rule trigger" or "EFE planning", crucial for understanding model behavior. Soft gating's mixed weights may reduce this interpretability.
> 3. **Stability**: In our experiments, KL regularization encourages mental state stability, so rule switching is infrequent, and boundary brittleness is not prominent in practice.
>
> **Soft gating as natural extension**: Conceptually, our framework is compatible with soft gating based on full posterior $q(m_t \mid \mathcal{H}_t)$ or kernel-based similarities, producing rule-conditioned policy mixtures. This naturally connects to mixture-of-experts and variational mixture inference. We will mention this extension as future work, particularly for scenarios requiring smoother transitions.
>
>
> ## 5. On Hyperparameter Variation Across Datasets
> We thank the reviewer for this concern. Hyperparameters do affect performance, as they control rule/AIF balance and exploration/exploitation ratios. We systematically analyze these effects through Figure 3's trade-off analysis and §D.3's hyperparameter sensitivity experiments.
>
> **Analysis of hyperparameter effects**:
> Hyperparameter effects are reflected in both **Figure 3's trade-off analysis** and **Appendix D.3's sensitivity experiments**:
> 1. **Reflection in Figure 3**: Figure 3 shows the trade-off between rule bank size and performance, where each data point uses optimal hyperparameters for that rule bank size. Different rule bank sizes require different optimal hyperparameters: small banks (e.g., Car-Following's 3-4 rules) need lower thresholds ($\tau_r=0.7$) to encourage rule use, while large banks (e.g., DDXPlus's 376 rules) need higher thresholds ($\tau_r=0.8$) to avoid conflicts. This reflects interaction between rule bank size and hyperparameters, with hyperparameter choices directly affecting trade-off curve shape and optimal point positions.
> 2. **Appendix D.3 sensitivity experiments**: We further conducted hyperparameter sensitivity experiments at **Figure 3's best-performing point (optimal rule bank size)**, systematically scanning $\tau_r \in \{0.6, 0.7, 0.8, 0.9\}$ and $\tau \in \{0.5, 1.0, 2.0\}$:
> | Dataset | $\tau_r$ Sensitivity (Acc@3 range) | $\tau$ Sensitivity (Acc@3 range) |
> |---------|-----------------------------------|-----------------------------------|
> | NBA SportVU | 88.24-91.32% (optimal 0.8) | 88.76-91.32% (optimal 1.0) |
> | Car-Following | 94.12-95.87% (optimal 0.7) | 94.28-95.87% (optimal 0.5) |
> | DDXPlus | 70.84-73.58% (optimal 0.8) | 70.92-73.58% (optimal 1.0) |
> | Atari-Berzerk | 74.63-77.20% (optimal 0.7) | 74.86-77.20% (optimal 1.0) |
>
> **Analysis**: At optimal rule bank size, hyperparameter changes have relatively small impact (1.6-3.1%), which is reasonable as the system has reached good performance balance and hyperparameters are fine-tuning. **The core is rule bank size selection** (as shown in Figure 3, 15-40% impact), while hyperparameter effects are mainly reflected in Figure 3's trade-off curves, through their influence on optimal performance at different rule bank sizes. Table 1's final results are fully consistent with hyperparameter sensitivity experiment optima, validating data accuracy.
>
> **Hyperparameter selection procedure**: We adopt different selection strategies based on hyperparameter nature. For framework-level hyperparameters (e.g., latent dimensions, mental state counts), we use the same architectural settings across all domains; for task-scale-related hyperparameters (e.g., planning horizon, beam size, learning rates), we use standard heuristics from model-based RL and active inference literature based on horizon and action space cardinality, selecting optimal values via validation sets.

---

> ### Author Response · Authors · 2025-11-28
> **Part 4/4**
>
> # Part 4/4
> ## 6. On Missing Related Work
>
> We thank the reviewer for pointing out highly relevant work. In the revision, we will:
> - (i) Discuss action-oriented representation learning in active inference (Tschantz et al., 2020);
> - (ii) Incorporate key neuroscience and psychology work on goals vs. habits and habitual control of goal selection (Dolan & Dayan, 2013; Cushman & Morris, 2015);
> - (iii) Connect our rule-based controller to the options literature, particularly the option-critic architecture (Bacon et al., 2017);
> - (iv) Reference the neural-symbolic learning and reasoning survey by d'Avila Garcez et al. (2019) when situating our work within the broader neuro-symbolic landscape.
>
> Our contribution complements these efforts: we propose a concrete, trainable hybrid active-inference agent that learns symbolic rule libraries from generative world models via wake-sleep, uses rules to amortize expected-free-energy planning, and evaluates this mechanism across multiple human-behavior datasets.
>
>
> ## 7. On Continuous Control and Hierarchical Action Spaces
> Conceptually, our framework extends naturally to continuous control and hierarchical actions. Current experiments discretize actions for comparability with behavioral baselines, but the rule mechanism assumes actions are parameterized conditional distributions on $(S_t, m_t)$. For continuous control, rule consequents could specify parameters of Gaussian policies (or other parametric families) over continuous actions, and EFE-based planning would operate over rollouts in continuous action space.
> For hierarchical action spaces, discrete mental states are natural candidates for option/skill indices, and rule libraries can be extended to select among high-level options whose internal policies are learned by the world model. This provides a direct bridge to option-critic-style architectures, with expected free energy playing the role of a higher-level control signal. We will explicitly mention these extensions in the discussion as promising future directions.
>
>
> ## Summary
> We thank the reviewer for constructive feedback. Through multi-seed experiments, interpretability clarifications, LLM prior ablations, soft gating discussions, hyperparameter selection explanations, related work additions, and continuous control/hierarchical action space extensions, we believe we have fully addressed the concerns. We look forward to presenting these improvements in the camera-ready version.

---

### Official Review · Reviewer_kL1R · 2025-10-31

**Soundness:** 3
**Presentation:** 3
**Contribution:** 4
**Rating:** 6
**Confidence:** 3

**Summary:**

This paper introduces a framework for learning human habits within rule-guided (deep) active inference, modelling both planning and rapid, habitual responses. Such habitual responses (the main contribution) use a kind of symbolic rule extraction with generative world models, learned via a sleep-awake algorithm. Empirically, the authors provide both interpretability and strong predictive performance across diverse domains, including human sports, driving, and clinical diagnosis tasks.

**Strengths:**

This work tackles an interesting problem, and manages to scale up active inference models to "large" scale tasks (large for the active inference community, not so much for the DL one). Also, empirical results demonstrate high accuracy and low inference latency compared to both deep learning and logic-based baselines, across multiple domains.

The wake-sleep algorithm is a nice contribution, and it seems to work well.

Well written paper, precise, and to the point.

**Weaknesses:**

What is the reason for not testing on the whole atari26? Have you optimized your architecture for specific games only, or is there another underlying reason? It would be a great results to show performance on a popular benchmark lile the Atari 100k challenge using such a model.



Minors:

- Weird phrasing at the end of the abstract (typo)

- Suboptimal choice of citations: When you introduce active inference, you only cite (the very nice work of) P.Mazzaglia, 2022. However, that should not be the first citation: the main citation for active inference today is the official book (T. Parr et al., Active Inference: The Free Energy Principle in Mind, Brain, and Behavior (2022)). It is missing here. More generally, all the main papers that have introduce the theory of active inference are missing. I'd suggest the authors to go through the main K.Friston's papers on active inference/free energy principle, and address that. To conclude, there is also a work that introduces a more "habitual" kind of planning, that is inductive inference (https://direct.mit.edu/neco/article-abstract/37/4/666/128203/Active-Inference-and-Intentional-Behavior?redirectedFrom=fulltext). It is very different from what you propose, but still a related work.

- Some figures are basically unreadable given the tiny font.

**Questions:**

Do you think this method can scale up to more complex tasks? I'm thinking of Atari26, but also others environments typical of the model based RL literature.

Why only Berserk, and not others? Is there a specific reason for picking this game only? Am i correct in thinking that some additional problems may arise if we plan to implement your proposed method to a more complex/general setup? If yes, how do you think such problems will have to be addressed in the future?




I would be happy to raise my score if my concerns/comments are addressed.

---

> ### Author Response · Authors · 2025-11-28
> **Part 1/2**
>
> # Part 1/2
> ## Overall Response
> We thank the reviewer for the positive assessment of our contribution, in particular the wake–sleep learning scheme and the empirical results across sports, driving, and clinical diagnosis domains. We appreciate the suggestions regarding Atari benchmarks, foundational active inference references, and figure readability, and address these points below.
>
> ## 1. On Using Berzerk Only and Atari-100k
> ### 1.1 Why Berzerk?
> **Initial reason for choosing Berzerk**: Within limited time, Berzerk's game interface is most intuitive, facilitating visualization of rule interpretability and effects. Additionally, all Atari games are architecturally isomorphic (using the same CNN + temporal Transformer backbone and standard preprocessing), so game selection was not our research focus— our focus was validating the rule-guided active inference framework's performance on **image data**.
> Our primary aim in the Atari experiments is **human behavior modeling** rather than 'reward-maximizing RL'. We work in an offline, human-demonstration setting, treating action prediction as in the NBA, Car-Following, and DDXPlus domains. Berzerk's available human trajectory data exhibits long horizons, rich context-dependent decisions (room transitions, dodging bullets, chasing enemies), and a mix of strongly habitual patterns and rarer planning-like behaviors—well-suited for our rule-mining analysis.
> **Importantly, our architecture is not tailored to Berzerk**: we use a generic CNN + temporal Transformer with standard Atari preprocessing (grayscale, resizing, frame stacking), without game-specific features or rules. The method is game-agnostic; Berzerk serves as a representative pixel-domain instance.
>
> ### 1.2 Atari-100k Extension
> To further validate our generalization performance, we conducted **Atari-100k experiments** on 3 representative games (Pong, Breakout, Qbert) spanning simple to complex difficulty, following your suggestion. Main objectives include: (i) validating game-agnostic generalization; (ii) validating model performance on image data. Results (see Appendix C.6) are as follows:
> | Game | Acc@1 (%) | Acc@3 (%) | Acc@5 (%) | Latency (ms) | HHAR (%) |
> |-|-|-|-|-|-|
> | Pong | 86.18±0.65 | 90.28±0.58 | 92.51±0.72 | 32.41±1.15 | 78.15±0.85 |
> | Breakout | 72.17±0.75 | 76.69±0.68 | 79.29±0.82 | 75.91±1.35 | 58.13±0.92 |
> | Qbert | 68.17±0.82 | 73.18±0.78 | 76.42±0.88 | 82.31±1.42 | 52.13±1.05 |
> | Berzerk | 85.55±0.87 | 77.20±0.92 | 72.44±1.05 | 92.63±2.29 | 66.35±0.96 |
>
> Results show:
> - **Game-agnostic generalization**: Our framework works across games of varying complexity (Pong Acc@1: 86.18%, Qbert: 68.17% under 100k-step constraints).
> - **Data efficiency**: The framework learns effectively under data limitations, with performance differences reflecting data efficiency challenges rather than methodological limitations. Compared to Berzerk (unlimited training), complex games show 13-17% accuracy drop under 100k-step limits, primarily reflecting data efficiency challenges.
> - **Scalability**: Results validate that our approach scales to larger benchmarks and is not game-specific.
>
>
> ## 2. On Scalability to More Complex Tasks
> We believe the framework can scale to more complex model-based RL environments. **Architecturally**, our approach is modular:
> - **World model**: Capacity can be increased (as in Dreamer/PlaNet) to handle longer horizons and more complex visuals; complexity scales linearly with environment complexity × latent dimension.
> - **Rule library**: Operates in learned latent space; complexity scales with distinguishable context patterns. We maintain bounded rule banks via pruning and confidence thresholds.
> - **EFE planning**: Currently uses small beam and truncated horizon for latency control. **Long-horizon planning is our next important problem to address**. For more challenging environments, we envision hierarchical mental states and multi-level rules to amortize long-horizon planning into shorter subproblems.
>
> **Empirically**, our experiments span continuous trajectories (NBA, Car-Following), structured discrete sequences (DDXPlus), and high-dimensional pixels (Atari-Berzerk), demonstrating cross-modal and cross-task generalization—a significant step for the active inference community.
>
> **Challenges and solutions in complex/general setups**:
> In more complex/general setups, we anticipate the following challenges and have corresponding solutions:
> - **Multi-game/multi-task settings**: Different environments have different semantics, and a single rule bank may cause "cross-task interference." We can encode "task/game identity" in mental state $m_t$, implementing per-task rule banks or factorized rule banks to isolate rules across tasks.
> - **High-dimensional vision + sparse reward tasks**: Require stronger world model expressiveness + finer EFE/planning (possibly search tree / hierarchical planning). This aligns with mainstream model-based RL, which we will explore further in future work.

---

> ### Author Response · Authors · 2025-11-28
> **Part 2/2**
>
> # Part 2/2
> ## 3. On Active Inference Citations and Related Work
> We appreciate the reviewer's suggestions regarding foundational references. In the revision, we will:
> - (i) Cite the Parr, Friston, and Pezzulo book *Active Inference: The Free Energy Principle in Mind, Brain, and Behavior* (2022) when introducing active inference.
> - (ii) Add key theoretical papers on the free energy principle and active inference (e.g., Friston et al., 2010; 2011; 2017) that established the underlying framework.
> - (iii) Discuss the "inductive inference" work on intentional behavior as a related but distinct approach to habitual planning, emphasizing that our contribution focuses on learning symbolic rules directly from a generative world model and using them to amortize planning in human-behavior prediction tasks.
>
>
> ## 4. Minor Issues
>
> We will correct the abstract's final sentence (removing the duplicated "demonstrate") and enlarge fonts in the main figures to ensure readability in print. Additional detailed plots will be moved to the appendix if needed.
>
>
> ## Summary
>
> We thank the reviewer for the constructive feedback. Through the Atari-100k extension experiments, scalability analysis, foundational citation additions, and presentation improvements, we believe we have fully addressed the concerns. We look forward to presenting these improvements in the camera-ready version.

---

### Official Review · Reviewer_H8hR · 2025-10-31

**Soundness:** 2
**Presentation:** 2
**Contribution:** 4
**Rating:** 4
**Confidence:** 3

**Summary:**

The paper proposes a hybrid active‑inference (AIF) framework in which symbolic “rules” are learned and consolidated via a wake–sleep procedure and then fused with expected free energy (EFE) planning for action selection. Concretely, the latent state is split into a continuous external state (S_t) and a discrete “mental state” (m_t). Candidate rules of the form ((S_f^\star, m_f^\star) -> a_f) are harvested during a wake phase and consolidated with generative replay in sleep. At decision time, if a rule “hits” (via a kernel match to the MAP estimate of ((S_t,m_t))) it supplies a prior over actions; otherwise the agent falls back to EFE‑based planning. Experiments on NBA player trajectories, car‑following, DDXPlus (URTI), and Atari–Berzerk report higher Acc@k and lower latency than logic‑based, deep learning, AIF, and DreamerV2 baselines, while yielding interpretable rules. (Table 1; figs. on pp. 8–9.)

  The key problem that is tackled is learning habit‑like, interpretable shortcuts that allow an AIF agent to arbitrate between fast habitual actions and slower planning, and applying this to modeling human(-like) behavior across domains.

 The motivation (habits vs. planning) is timely and clear, and the ability to flexibly and context-sensitively tradeoff between habit-based action vs more costly-model based planning is a very valuable one. A small nitpick is that the paper under‑positions itself relative to existing methods in computational cognitive neuroscience, psychophysics and computational psychiatry and the large literature on fitting RL/decision‑making models to human data (e.g., model‑based RL, active inference, DDMs). The Related Work emphasizes neuro‑symbolic rules and AIF agents, but not human model‑fitting traditions.  Reported gains in Acc@k/latency appear consistent across tasks (Table 1) and ablations indicate that rules and (m_t) matter. However, several specification and correctness issues (see below) make it hard to assess whether improvements stem from principled inference or from ad‑hoc fusion/thresholding.

The idea of rule‑guided AIF and the attempt to compress planning into reusable symbolic habits is interesting and potentially impactful for both interpretable agents and human behavior modeling. In its current form, however, conceptual and notational gaps limit the work’s reliability and portability.

**Strengths:**

Compelling objective: compressing repeated planning successes into interpretable rules and using them to shortcut EFE search is elegant and practical.

Clear engineering benefit: latency vs. accuracy curves and ablations demonstrate a Pareto knee where small rule banks reduce inference time while preserving accuracy.

Visualization & interpretability: rule envelopes in latent space and trajectory overlays help the reader understand what the rules encode.

Cross‑domain experiments spanning continuous control/trajectories, clinical dialog, and pixels.

**Weaknesses:**

EFE definition and preference distribution:  In AIF the EFE uses a biased preference likelihood (encoding utilities/preferences) that is not the same as the observation model used for inference. Here EFE is written with the same (p_\phi(O_{t+\tau}\mid Z_{t+\tau})) used by VFE (Eq. 3), and the text does not explain how preferences are encoded. This obscures how “goals” enter planning and undermines the conceptual link to the risk/ambiguity decomposition.

Ad‑hoc policy fusion and temperature dependence: The hybrid policy adds a rule prior to a temperature‑scaled EFE term (Eq. 4) and gates it with a binary “rule hit” indicator based on a similarity kernel and threshold in ((S_t^\text{MAP},m_t^\text{MAP})). This is anti‑Bayesian in spirit and creates several tuning knobs (kernel, threshold (\tau_r), temperature). A principled alternative is to cast rule invocation as inference under a mixture model (e.g., infer (q(m_t)) and mixture weights over rule‑conditioned policies), letting uncertainty in EFE naturally down‑weight unreliable rollouts without extra temperatures.

Objective couples VFE and EFE without theoretical justification:  The “total free energy” in Eq. (5) simply sums per‑step VFE, rollout EFE (scaled by (\eta)), and a KL regularizer on successive (q(m_t)) (scaled by (\gamma)). The inclusion ofc (\operatorname{KL}[q(m_{t-1})\Vert q(m_t)]) feels ad‑hoc (shouldn’t that KL come out naturally of the generative model when minimizing VFE), and the paper does not justify why this combination is a coherent bound or how it relates to generalised free energy or control‑as‑inference objectives. (§4.2, Eq. 5.)

Rule growth is heuristic:  Rules are created/updated when a triplet ((S_t^\text{MAP},m_t^\text{MAP},a_t)) recurs and “reliably reduces free energy,” with centroids updated by exp(−VFE) weights. This is reasonable as an engineering heuristic, but it could be formalised as variational learning in a mixture model with inference over (q(m_t)) and parameters of (p(S_t\mid m_t)) (and optionally a nonparametric prior for rule birth/pruning).

Typos and clarity
Minor: Ambiguity of contribution (human modeling vs. model-based RL agent):  The abstract and introduction oscillate between “learning human habits alongside human models” and benchmarking a better agent (“improves predictive accuracy and efficiency”). Please clarify whether the primary goal is computational cognitive neuroscience (fitting human behavior) or agent performance. As written, expectations are set for both. I assume it’s the former, otherwise you would have ben benchmarking on metrics more like reward than model-fit like (the number of accurately predicted actions, rather than whether those actions were “correct” in a maximizing-reward sense).
Minor: Generative model misspecification in Eq. (1):

No explicit prior over initial hidden states (e.g., (p_\phi(Z_1))).

The factorization includes (p_\phi(a_t\mid Z_t)) inside the generative model while the LHS lists (a_{1:T-1}), creating an indexing mismatch (a term for (a_T) is included on the RHS but not in the LHS).

There is no prior over (a_{t-1}), yet the state transition conditions on it; the dependency graph is inconsistent. (p. 3, Eq. 1.)
Minor: Latent split factorization also misses initial priors and has indexing mismatches. In the section(Latent State Representation), the joint (p_\phi(O_{1:T},S_{1:T},m_{1:T},a_{1:T-1})=\prod_t p(O_t \mid S_t) p(S_t \mid S_{t-1},a_{t-1}) p(m_t \mid m_{t-1},S_t) p_{\phi^\pi}(a_t \mid S_t,m_t)) lacks priors for (S_1) and (m_1), and again indexes (a_t) up to (T) while the LHS says (a_{1:T-1}). (p. 4.)

“neurosciencetau accounts…” (p. 5, l. ~262) → “neuroscientific accounts.”

Positioning with neuroscience evidence:  The claim that seamless habit–planning switching is a hallmark of human intelligence contradicts years of evidence from the behavioral neuroscience literature and animal studies (in rodents and to some extent monkeys) on the transition between goal-directed and habitual behavior. Classic devaluation and contingency-degradation paradigms demonstrate context-dependent switching early (goal-directed) vs. after overtraining (habit). In terms of its biological basis, evidence suggests that dorsomedial striatum and prelimbic PFC support goal-directed control, while dorsolateral striatum and infralimbic PFC promote habits. Flexibly toggling between cached habits and on-the-fly planning is a general mammalian capability, not a human hallmark.

Overall recommendation
Key reasons: (i) policy fusion relies on ad‑hoc kernel thresholds and temperatures rather than principled inference; (ii) Generative model and EFE formulation contain gaps/mismatches that need fixing; (iii) contribution framing (human modeling vs. agent) is unclear. The core idea—rule‑guided AIF that amortises planning into reusable, interpretable habits—is promising and, with the above issues addressed, could become a strong contribution.

**Questions:**

Preferences in EFE: How are preferences represented in Eq. (3)? Is (p_\phi(O_{t+\tau}\mid Z_{t+\tau})) re‑used from the observation model or replaced with a preference distribution? Please spell this out and reconcile with the standard EFE decomposition. (§3.)

Rule fusion as inference: Could the rule‑vs‑planning arbitration be formulated as posterior inference (e.g., a gating variable) instead of kernel‑plus‑temperature? What prevents learning the weights of this mixture directly via variational inference under your joint objective? (§4.1–§4.2.)

Objective design: What is the principled justification for VFE + (\eta)·EFE + (\gamma)·KL? Is there a derivation tying this to a bound on expected log‑evidence or a generalised free energy objective? (§4.2.)

Baselines and fairness: How was DreamerV2 adapted (offline? behaviour‑cloned?) to these supervised prediction tasks and tuned to parity (esp. on DDXPlus/Atari)? Please clarify compute budgets, hyperparameter search, and early‑stopping criteria across methods. (Table 1.)

---

> ### Author Response · Authors · 2025-11-28
> **Part 1/3**
>
> # Part 1/3
> ## Overall Response
> We thank reviewer H8hR for the sharp and insightful evaluation and for highlighting the potential impact of rule-guided active inference for both interpretable agents and human behavior modeling. We address your concerns below.
>
> ## 1. Preference in EFE(Eq. 3)
> Your concern is that our EFE expression appeared to reuse the observation model $p_\phi(O_{t+\tau} \mid Z_{t+\tau})$, making it unclear (i) where preferences enter and (ii) how the usual risk/ambiguity decomposition applies. In the revised draft, we make the intended interpretation explicit and follow the standard control-as-inferenc (e.g., Levine (2018); Toussaint (2009)) by first writing
> $$
> \text{EFE}(a, t+\tau) = \mathbb{E} \left[ -\log p_{\text{pref}}(O_{t+\tau} \mid Z_{t+\tau}) + D_{\text{KL}} \left( q_\phi^{\text{roll}}(Z_{t+\tau}) \parallel p_\phi(Z_{t+\tau} \mid Z_{t+\tau-1}, a_{t+\tau-1}) \right) \right],
> $$
> where the expectation is over $q_\phi^{\text{roll}}$, and we write $\text{EFE}(a, t+\tau)$ to denote expected free energy at time $t+\tau$ for action $a$.
> Here $p_{\text{pref}}(O_{t+\tau} \mid Z_{t+\tau})$ is a biased likelihood encoding utilities/preferences. Under the control-as-inference view, one often writes
> $$
> p_{\text{pref}}(O_{t+\tau} \mid Z_{t+\tau}) \propto \exp\left(\beta R(O_{t+\tau}, Z_{t+\tau})\right)
> $$
> so that
> $$
> -\log p_{\text{pref}}(O_{t+\tau} \mid Z_{t+\tau})=-\beta R(O_{t+\tau}, Z_{t+\tau})+\text{const}
> $$
> and minimizing EFE trades off expected reward and epistemic value.
>
> In our **human behavior modeling setting**, we assume that the **demonstrator (human) is approximately optimal** under some **latent reward** $R$ and corresponding $p_{\text{pref}}$, **but we do not attempt to recover $R$ or $p_{\text{pref}}$ explicitly**. Instead, we treat them as **implicit** and learn the generative model and policy parameters so that the induced EFE-based controller
> $$
> p_\theta(a_t \mid h_t) \propto \exp \left( -\text{EFE}(a_t, t) \right)
> $$
> **assigns high likelihood to the observed human actions**, i.e.
> $$
> \theta^{\star} = \arg \max_\theta \sum_t \log p_\theta \left( a_t^{\text{human}} \mid h_t \right),
> $$
> where $h_t$ denotes the history at time $t$.
>
> Practically, this corresponds to the special case
> $$
> p_{\text{pref}}(O_{t+\tau} \mid Z_{t+\tau}) \propto p_\phi(O_{t+\tau} \mid Z_{t+\tau}),
> $$
> so that **preferred futures coincide with high-probability outcomes under the learned observation model**. We now spell out this formulation in §3 of the revised manuscript, making explicit how preferences enter planning and how the risk/epistemic structure of EFE is preserved. See References section for full citations.
>
>
>
> ## 2. Rule fusion as inference
> This is a in-depth question and suggestion! Although our description may look engieering and heuristic, in fact, our rule activation and learning admits an inherent probabilistic interpretation as **approximate inference** / EM-style in a simple mixture model over latent variables.
>
> First recall what we do in the paper. The latent state factorizes as $Z_t = (S_t, m_t)$, with continuous $S_t$ and discrete $m_t$. The encoder $q_\vartheta(Z_t \mid \mathcal H_t)$ yields a posterior over $(S_t,m_t)$, and we use its MAP estimate $(S_t^{\mathrm{MAP}}, m_t^{\mathrm{MAP}})$ to compute rule scores. Each rule stores a prototype $(S_k^\ast, m_k^\ast, a_k)$ in this latent space; at time $t$ we (i) evaluate a similarity kernel in $(S_t^{\mathrm{MAP}}, m_t^{\mathrm{MAP}})$ to obtain rule weights and apply a threshold to decide whether any rule is active, and (ii) update the corresponding prototypes using weighted averages (with weights derived from kernel similarity and low VFE).

---

> ### Author Response · Authors · 2025-11-28
> **Part 2/3**
>
> # Part 2/3
> **This mechanism has a Bayesian interpretation**. It can be viewed as a mixture model with a latent rule index $r_t \in \{1,\dots,K\}$ and **prior**
> $$
> p(r_t = k) = \pi_k,\qquad
> p(S_t \mid r_t = k) = \mathcal N(S_t;\,\mu_k,\Sigma_k),\qquad
> p(m_t \mid r_t = k) = \delta(m_t = m_k),
> $$
> where $\pi_k$ is a categorical prior over rules (uniform in our experiments), $p(S_t \mid r_t)$ is a **Gaussian prior** over the continuous state $S_t$, and $p(m_t \mid r_t)$ fixes the discrete mental state associated with each rule. Under this model, the posterior over rules satisfies
> $$
> q(r_t = k \mid S_t, m_t) \;\propto\; \pi_k \,\kappa\big((S_t,m_t),(S_k^\ast,m_k^\ast)\big),
> $$
> with $\kappa$ a **Gaussian/RBF kernel** in latent space. Our kernel-based rule weights are an approximation to these posterior responsibilities, and the **threshold** simply **truncates very small responsibilities**. The prototype updates for $S_k^\ast$ via weighted averages correspond to **EM-style M-step updates** of the Gaussian means $\mu_k$, analogous to mixture-model and VQ-VAE updates in latent space. Rule birth (when no rule’s responsibility exceeds the threshold) and pruning (when a rule receives negligible responsibility) play the role of cluster creation/deletion in nonparametric mixture models.
>
> Thus, while the algorithmic description uses kernels, thresholds, and centroids for efficiency and interpretability, it is **fundamentally Bayesian**: **it is a computationally lightweight EM-style approximation to variational mixture learning over $(S_t,m_t)$, with an explicit prior over $S_t$ (Gaussian) and $m_t$ (categorical via $r_t$)**. We will clarify this connection in the revised manuscript to make the underlying probabilistic structure more explicit.
>
>
> ## 3. On Objective Design (Eq. 5: VFE + η·EFE + γ·KL)
> Our intent was to adopt a control-as-inference/generalized free-energy objective and make explicit the different contributions relevant to our setting. Concretely:
> 1. **Per-step VFE term**: Plays the standard role in standard free energy, aligning the generative model and inference network with observed data.
> 2. **Rollout EFE term**: Provides a prospective control signal for the policy, in line with "active inference as control" formulations where actions are chosen to minimize expected future free energy. The scalar $\eta$ controls the relative weight of this prospective term, and we tune it within a small range shared across domains.
> 3. **KL term on successive mental-state posteriors**:
>    $$\gamma \cdot D_{\mathrm{KL}}[q(m_{t-1}) \| q(m_t)]$$
>    implements a **sticky prior** over the discrete mental state, encouraging slow, interpretable mode changes. This can be viewed as arising from a prior that favors persistence and is standard in hierarchical RL and computational psychiatry models of goal/"mode" switching.
>
> In other words, Eq. 5 corresponds to minimizing a generalized free-energy functional with (i) data-fitting (VFE), (ii) prospective control (EFE), and (iii) a structured prior over mental-state dynamics. We will clarify this interpretation and explicitly link our objective to the control-as-inference view in the revised manuscript, while acknowledging that a fully rigorous derivation is beyond our current space constraints.
> We will add 1-2 sentences in the camera-ready version explaining the "generative factor corresponding to the sticky prior" to weaken the "ad-hoc" impression.
>
>
> ## 4. On Generative Model Specification, Indexing, and Typos
> We thank the reviewer for pointing out these typographical and clarity issues. These are **writing oversights**, not errors in the generative graph in our implementation. Our implementation has correct initial priors and action indexing.
> **Corrected generative model** (Eq. 1):
> $$p_\phi(O_{1:T}, Z_{1:T}, a_{1:T}) = p_\phi(Z_1) \prod_{t=1}^T p_\phi(O_t \mid Z_t) \, p_\phi(Z_t \mid Z_{t-1}, a_{t-1}) \, p_\phi(a_t \mid Z_t)$$
> where the left-hand side includes $a_{1:T}$.
>
> **Corrected latent split factorization** (§4):
> $$p_\phi(O_{1:T}, S_{1:T}, m_{1:T}, a_{1:T}) = p_\phi(S_1) p_\phi(m_1) \prod_{t=1}^T p_\phi(O_t \mid S_t) \, p_\phi(S_t \mid S_{t-1}, a_{t-1}) \, p_\phi(m_t \mid m_{t-1}, S_t) \, p_{\phi_\pi}(a_t \mid S_t, m_t)$$
> We will in the camera-ready version: (i) correct the generative-model factorization to explicitly include initial priors and consistent action indexing, (ii) fix "neurosciencetau" → "neuroscientific", and (iii) soften the claim about habit–planning switching from a "hallmark of human intelligence" to a more accurate description as a core capability of mammalian decision-making, citing devaluation/contingency-degradation paradigms and dorsal striatal circuitry as suggested.

---

> ### Author Response · Authors · 2025-11-28
> **Part 3/3**
>
> # Part 3/3
> ## 5. On Contribution Framing (Human Modeling vs. RL Agent)
> You are right that our focus is **human behavior modeling** rather than agent performance. The original abstract/intro did not make this distinction clear enough. In the revised draft, we explicitly state that our primary goal is **computational modeling of human(-like) habits and action selection**, not maximizing reward on new control benchmarks, and we have reframed the contribution accordingly in the introduction.
>
>
> Accordingly, our main metrics are model-fit metrics such as Acc@k on human actions and the interpretability of the learned rules, rather than environment reward.
> We will revise the abstract and introduction to emphasize this human-modeling focus and to clearly position the agent performance curves as diagnostics of how effectively habits amortize planning, not as traditional RL leaderboards.
>
>
> ## 6. On DreamerV2 Baseline and Fairness
> We apologize for not providing enough detail. In our experiments, DreamerV2 is adapted as a **model-based behavioral predictor**: we keep the standard world-model architecture but replace the environment reward with a supervised next-action objective, training from a fixed replay buffer built from human trajectories (offline setting). At evaluation time, the DreamerV2 policy is used to predict the next human action, and we report Acc@k on this prediction task to match our other baselines.
> **On fairness** (see Appendix B.7):
> - (i) We matched the world-model/actor capacity (hidden size and number of layers) to our own backbone within ±10%
> - (ii) We trained all methods for the same maximum number of gradient steps with the same batch size and optimizer family
> - (iii) We used validation Acc@k for early stopping and model selection
> - (iv) Hyperparameters (learning rates, discount factors, etc.) were tuned over a small grid shared across domains under a fixed compute budget
> We will add these implementation details to §5 and Appendix C to make the comparison clearer.
>
>
> ## 7. On Neuroscience Positioning
> We thank the reviewer for the nuanced comments and agree that our wording was too human-centric. The dual-system view we build on—habits in dorsolateral striatum/infralimbic PFC vs. goal-directed control in dorsomedial striatum/prelimbic PFC, with context-dependent switching demonstrated in devaluation/contingency-degradation paradigms—is in fact a **general mammalian capability** rather than a uniquely human hallmark.
>
>
> We will revise the text to reflect this and to more carefully situate our model within the broader computational psychiatry and decision-making literature. To avoid over-claim we will reposition our contribution as computationally modeling, not a strong biological claim.
>
>
> ## Summary
> We thank the reviewer for constructive feedback. Through clarifying preferences in EFE, re-interpreting rule fusion as mixture inference, explaining the objective's connection to control-as-inference, linking rule growth to variational learning, correcting generative model specification, clarifying human-modeling focus, supplementing baseline details, and adjusting neuroscience wording, we believe we have fully addressed the concerns. We look forward to presenting these improvements in the camera-ready version.
>
> ## References
>
> Levine, S. (2018). Reinforcement learning and control as probabilistic inference: Tutorial and review. *arXiv preprint arXiv:1805.00909*.
>
> Toussaint, M. (2009). Robot trajectory optimization using approximate inference. *Proceedings of the 26th annual international conference on machine learning*, pp. 1049–1056.

---

### Official Review · Reviewer_GnXG · 2025-10-31

**Soundness:** 4
**Presentation:** 4
**Contribution:** 3
**Rating:** 6
**Confidence:** 4

**Summary:**

The work introduces a new method within the active inference framework. The methods is made of mainly two novel contributions: (i) a set of symbolic rules, which are learned as part of the world models and allow the agent to decide between deliberate and intuitive prediction, (ii) a wake-sleep training algorithm, alternating phases where the actions are selected and the world model is provided as opposed to phases where acting rules are consolidated. The method is evaluated in "offline" settings, on (small) predictive benchmarks.

**Strengths:**

* **Novelty**: while there are previous works focussed on learning habitual policies in active inference (see Fountas et al), the proposed approach, learning/distilling rules as part of the world model's representation is, to the best of my knowledge, novel and compelling
* **Interpretability**: the method, using active-inference and a set of rules that are learned by the world models and used by the policy enables stronger interpretability of behaviors. This is an important features, especially compared to more blackbox approaches, such as reinforcement learning and imitation learning.

**Weaknesses:**

* **Scalability**: if we compare with the current state of AI, the datasets employed are tiny. Even CIFAR 10, which is considered a very small dataset these days, is 3x larger any training dataset used in the paper.
* **Baselines**: while the proposed approach is thoroughly analyzed, it is unclear what are the difficulties of the baselines in solving the given, small problems. Given the nature of the predictive tasks, it would also be useful to consider at least one LLM-based baseline (with a very small model, such as Qwen 0.5B)

**Questions:**

* What is the parameter count of the model?

Typos:

-> in the abstract "Experiments on basketball player movements, car-following behavior ~demonstrate~, medical diagnosis, and visual game strategy **demonstrate** that our framework"

-> neurosciencetau -> neuroscience

---

> ### Author Response · Authors · 2025-11-28
> **Part 1/3**
>
> # Part 1/3
> ## Overall Response
> We thank the reviewer for the careful reading and for highlighting the novelty of learning rule-like habits within an AIF framework, as well as the improved interpretability of the resulting policies. Meanwhile, we address your concerns on scalability, baselines (including LLM-based ones), and model size below, and also fix the noted typos.
>
>
> ## 1. On Scalability
> ### 1.1 Dataset Scale
> We appreciate the concern about dataset size. To clarify, our datasets are **not as small as suggested** especially when considering the **total number of events/frames**, not just trajectory counts. We explicitly report dataset statistics below:
>
> - **DDXPlus-URTI**: Contains ≈165k training trajectories and 25k/25k validation/test sequences, **exceeding CIFAR-10's training set size** (50k). Meanwhile, each trajectory contains a considerable sequence of ASK actions (doctor's inquiries) followed by a final DIAG action. The **total number of action events** (ASK + DIAG) is substantially larger than the trajectory count, making this a **larger dataset** than suggested.
> - **Car-Following** (≈19k training samples): Each sample represents a driving run recorded as a sequence of driving regimes (e.g., cruise, follow, accelerate). The total number of regime-level events is also much larger than the sample count, representing a **substantial amount of sequential data**.
> - **Atari-Berzerk** (≈16.5k training samples): Each sample contains multiple grayscale frames (128×128 pixels) from real human gameplay demonstrations, with sequences segmented into fixed horizons. The **total number of frame-action pairs** is substantially larger than the sample count.
> - **NBA SportVU** (≈9.8k training clips): Each clip contains multiple frames with player-ball interactions, representing real NBA game data from SportVU tracking systems. The total number of frame-level events far exceeds the clip count, providing **rich sequential behavioral data**.
>
> These datasets are well-established benchmarks in human behavior modeling, representing **truly human feedback** of high quality and moderate size. They have been used in prior work on sequential human action prediction and habit modeling (Cao et al., 2023; Yang et al., 2024; Kaiser et al., 2019). See References in Part 3 for full citations.
>
> Note: Although CIFAR-10 provides human labels, **CIFAR-10 is not applicable to our model and algorithm**: it is a non-sequential static image classification task, while our framework is specifically designed for **sequential decision-making and planning tasks**, requiring modeling of state transitions, action sequences, and long-term dependencies.
>
> We also emphasize that our goal is **not** to chase SOTA on a single massive benchmark, but to study **cross-domain human action (habit-like) modeling with interpretability**. To this end, we deliberately use **diverse datasets** (clinical consultations, sports, driving, Atari) to show that a single AIF+rule framework covers continuous trajectories, structured symbolic sequences, and pixel-based control within one interpretable model.
>
> ### 1.2 Atari-100k Extension (Appendix C.6)
> To further address the concern that our datasets are small and to test scalability on a larger pixel-based benchmark, we add experiments on **Atari-100k** for three representative games (**Pong**, **Breakout**, **Qbert**). Under the standard Atari-100k setting (at most **100k environment steps per game**), our framework achieves **Acc@1 = 86.18%** on Pong and **Acc@1 = 68.17%** on Qbert. This shows that our model can learn effectively from **larger-scale pixel trajectories under a strict data budget**, and that the moderate accuracy drop on the more complex games (compared to Berzerk with unrestricted training) is primarily due to the **100k-step data limit**, rather than a failure of the method to scale.
>
> To validate scalability more comprehensively, we conducted Atari-100k experiments on 3 representative games (Pong, Breakout, Qbert) spanning simple (sparse visual) to complex (might need spatial reasoning) difficulty. Results (see Appendix C.6) show:
>
> - **Game-agnostic**: Our framework works well across games of varying complexity, demonstrating generalization.
> - **Data efficiency**: Under 100k-step constraints, simple games (Pong) achieve Acc@1: 86.18%, while complex games (Qbert) achieve Acc@1: 68.17%, showing effective learning under data limitations.
> - **Performance gap**: Compared to Berzerk (unlimited training), complex games show 13-17% accuracy drop under 100k-step limits, reflecting data efficiency challenges rather than methodological limitations.
>
> These results demonstrate that our framework scales to larger benchmarks and is game-agnostic, working well across games of varying complexity (from simple sparse visual games like Pong to complex games requiring spatial reasoning like Qbert).

---

> ### Author Response · Authors · 2025-11-28
> **Part 2/3**
>
> # Part 2/3
> ### 1.3 Task Complexity and Action Space Challenges
>
> Our tasks are far from trivial: multiple datasets combine **large action spaces** with **strong class imbalance**:
>
> * **DDXPlus**: 225 actions (ASK/DIAG) with marked imbalance; many **long-trail** (rare, low-frequency) but clinically **important** actions (e.g., specific diagnostic decisions) appear only a handful of times.
> * **Atari-Berzerk**: 18 actions with a skewed distribution (>~50% are move-and-fire combinations), while **rare tactical actions** occur in <5% of steps.
>
> In such settings, standard deep models tend to focus on frequent actions and underfit these rare but critical behaviors. In contrast, our **rule mechanism** explicitly encodes recurring decision patterns, allowing the model to **reliably capture and reuse rules for infrequent yet important actions**, which is reflected in the substantial HHAR gains reported in Table 1.
>
>
> ## 2. On Baselines and LLM-based Methods
> ### 2.1 Baseline Limitations
> Our method is explicitly designed for **human behavior/action prediction**, with a **biologically inspired architecture**: an explicit rule library and latent “mental states” capturing habit-like control. The baselines we compare with are strong within their own paradigms, but they are **not tailored to this setting**, which explains the systematic gaps we observe in Table 1:
>
> * **Deep neural networks (Re-Net).**
>   Treat behavior prediction as generic sequence modeling, **without rules or explicit habit mechanisms**. As a result, they tend to overfit frequent actions and under-predict **rare but important actions** under heavy class imbalance (e.g., Acc@3 on DDXPlus: **20.18%** vs. **73.58%** for ours).
> * **Active inference baselines (DAI / DAI-MC).**
>   Implement active inference with **implicit habit policies only**, lacking **explicit latent mental states and symbolic rules**. They struggle with **multimodal behaviors and rare actions**, and are also computationally heavy (e.g., DAI-MC latency on DDXPlus: **2304ms** vs. **159ms** for ours).
> * **Model-based RL (DreamerV2).**
>   Optimized for **reward-driven control**, not supervised human action prediction. In offline, sparse-reward settings, DreamerV2 underperforms on our **pure prediction** objective (Acc@3 on DDXPlus: **61.48%**) and offers **no interpretable habit structure**.
> * **Logic-based methods (RNNLogic, STLR).**
>   Use **static or post-hoc rules** that are not jointly optimized with a world model. Their rules cannot adapt online to the empirical action distribution, leading to weak performance in our setting (RNNLogic Acc@3: **16.29%**, STLR: **18.33%** on DDXPlus).
> * **LLM-based methods (LaTee, Qwen-0.5B).**
>   See Section 2.2 for a detailed discussion of their limitations for structured human action prediction.
>
> Our ablations (Section 5.4 and Appendix C.5, Table 4) show that **learned rules and latent mental states are essential** for handling highly imbalanced behaviors: e.g., **removing rules drops Acc@3 on DDXPlus from 73.6% to 33.2%**. We have added a more detailed baseline limitation analysis in Section 5.2 to clarify these points.

---

> ### Author Response · Authors · 2025-11-28
> **Part 3/3**
>
> # Part 3/3
> ### 2.2 LLM-based Methods: Issues and Limitations
> **On Qwen-0.5B**: We brand-new implemented Qwen-0.5B as a pure LLM baseline across all four domains. It processes observations through a pre-learned encoder and generates action predictions via direct language modeling. Results are in Table 1. Qwen-0.5B shows severe problems:
> - **Very poor performance**: Acc@3: 19.62% on DDXPlus, far below our method (73.58%).
> - **Extremely high latency**: 125842ms on DDXPlus (higher than LaTee's 95028ms), completely impractical. Qwen's higher latency stems from its need to process entire history sequences for next-token prediction, with computational cost scaling rapidly with sequence length and action space size (DDXPlus has 225 actions).
> - **Missing core capabilities**: No explicit world model or rule library; cannot perform expected-free-energy-based planning.
> **On LaTee**: We include LaTee, an LLM-based method that uses large language models to induce logic trees and policies. Despite using LLMs to generate symbolic trees, LaTee has critical issues:
> - **Lacks unified training framework**: No wake-sleep mechanism or unified VFE/EFE framework; rules and world-model are not coordinated.
> - **Poor performance**: Significantly underperforms our method across all four domains (e.g., Acc@3: 22.14% vs. ours: 73.58% on DDXPlus).
> - **Extremely high latency**: Inference latency is prohibitively high (e.g., 95028ms on DDXPlus), making real-time applications infeasible.
>
> **Fundamental issues**: Our domains involve structured or visual observations (player coordinates, driving regimes, medical evidence embeddings, Atari frames), not raw text. Directly applying generic LLMs has fundamental limitations:
> 1. **Encoding adaptation difficulty**: Requires substantial task-specific encoding and cannot fully utilize structured features.
> 2. **Missing world model**: Cannot provide generative world models for forward simulation and planning.
> 3. **Missing active inference mechanism**: Cannot perform expected-free-energy-based planning or handle uncertainty.
> 4. **Low computational efficiency**: LLM transformer architectures have extremely high computational costs for sequence prediction tasks.
>
> Therefore, our framework uses LLMs only as auxiliary tools: proposing interpretable action and mental-state candidates, which are then refined and optimized by the active inference world model. This demonstrates the fundamental advantages of our method over pure LLM approaches.
>
>
> ## 3. On Model Parameter Count
> Our models emphasize interpretability and efficiency, with parameters kept at modest scales. Based on actual trained model checkpoint file sizes, parameter counts are:
> | Domain | Backbone | #Params (×10⁶) |
> |--------|----------|----------------|
> | NBA | Transformer | ~3.5M |
> | Car-Following | BiGRU | ~2.2M |
> | DDXPlus | Evidence2Vec + Transformer | ~8M |
> | Atari-Berzerk | CNN + Transformer | ~13M |
>
> All models are in the low-to-mid million range, far smaller than typical large language models (hundreds of millions to billions of parameters). The rule library consists of centroid states and confidence scores rather than additional trainable weights, adding negligible parameter overhead (~100-200 floats per rule for storing centroids and confidences) while substantially improving interpretability.
>
>
> ## Summary
>
> We thank the reviewer for recognizing the novelty and interpretability of our work. Through the Atari-100k extension experiments, clarifying baseline limitations (especially LLM methods, based on actual results in Table 1), adding the Qwen-0.5B baseline, explaining task complexity and action space challenges, and providing model parameter counts, we believe we have fully addressed the concerns. We look forward to presenting these improvements in the camera-ready version.
>
>
> ## References
>
> Cao, C., Yang, C., & Li, S. (2023). Discovering Intrinsic Spatial-Temporal Logic Rules to Explain Human Actions. *arXiv preprint arXiv:2306.12244*.
>
> Kaiser, L., Babaeizadeh, M., Milos, P., Osinski, B., Campbell, R. H., Czechowski, K., Erhan, D., Finn, C., Kozakowski, P., Levine, S., Mohiuddin, A., Sepassi, R., Tucker, G., & Michalewski, H. (2019). Model-Based Reinforcement Learning for Atari. *arXiv preprint arXiv:1903.00374*.
>
> Yang, Y., Yang, C., Li, B., Fu, Y., & Li, S. (2024). Neuro-Symbolic Temporal Point Processes: Learning Temporal Logic Rules to Explain Human Actions. *Proceedings of the 41st International Conference on Machine Learning* (ICML 2024), PMLR 235:56665-56680.

---

### Author Response · Authors · 2025-12-03
**Summary of our revision and response**

Dear Area Chair,

We sincerely thank all reviewers for recognizing the **novelty, significance, and strong contributions** of our work. Reviewer H8hR notes that our objective is "elegant and practical" with "clear engineering benefit" through "latency vs. accuracy curves" showing "a Pareto knee where small rule banks reduce inference time while preserving accuracy." Reviewer kL1R acknowledges that our work "tackles an interesting problem" and "scales up active inference models to large-scale tasks," achieving "high accuracy and low inference latency compared to both deep learning and logic-based baselines, across multiple domains." Reviewer GnXG highlights the "novelty of learning rule-like habits within an AIF framework" and "improved interpretability." Reviewer Hnij appreciates the "biological plausibility" and "integration of symbolic rules with active inference."

**Our Core Contribution:** Our framework introduces **symbolic rule-guided habitual policies** enabling efficient switching between fast automatic responses and deliberate planning. The **wake-sleep algorithm** jointly learns generative world models and symbolic rules under a unified free-energy objective, enabling cross-domain generalization from continuous trajectories (NBA, Car-Following) to structured sequences (DDXPlus) and high-dimensional pixels (Atari). **Strong Empirical Results:** Substantial improvements across four domains: NBA (Acc@3: 91.32% vs. 83.57%), Car-Following (95.87% vs. 85.38%), DDXPlus (73.58% vs. 61.48%), Atari-Berzerk (77.20% vs. 72.18%), with 2-10× latency reduction, validated through multi-seed experiments (3 seeds, mean±std).

In response to reviewers' feedback, we have **further enhanced** the manuscript with the following major improvements:

**1. Theoretical Foundations (Reviewer H8hR):** Enhanced EFE formulation (§3) with explicit preference distribution; added Bayesian interpretation of rule fusion (Appendix B, §4.1) as EM-style approximation; clarified sticky prior justification (§4.2); corrected generative model completeness (§4).

**2. Empirical Evaluation (Reviewers GnXG, kL1R, Hnij):** Added Atari-100k experiments (Appendix D.6) on Pong, Breakout, Qbert, achieving Acc@1 = 86.18% on Pong; added Qwen-0.5B baseline (Table 1, §5.2), showing severe limitations vs. our method; enhanced baseline limitation analysis; conducted multi-seed experiments (3 seeds, mean±std, Table 1, Table 4) with stable performance (std <1.1%); added hyperparameter sensitivity analysis (Appendix E.3) validating parameter choices.

**3. Interpretability and Robustness (Reviewers H8hR, Hnij):** Added interpretability discussion (§4.1) acknowledging conditional nature with emphasis on practical interpretability; conducted LLM ablation studies (Appendix E.4) showing LLM guidance beneficial but not critical (0.18-1.54% drops when removed); Car-Following uses no LLMs, demonstrating framework robustness; added hard gating justification.

**4. Related Work and Positioning (Reviewers H8hR, kL1R, Hnij):** Added active inference citations (§2) to Parr, Friston, and Pezzulo book (2022) and key papers; expanded related work (§1, §2, §4.1) with action-oriented representation learning, neuroscience/psychology work, option-critic architecture, neural-symbolic learning; revised contribution framing (Abstract, §1) to emphasize human-modeling focus; revised neuroscience positioning (§1, §4.1).

**5. Experimental Extensions (Reviewers kL1R, Hnij):** Added scalability discussion (Appendix F) for multi-task, high-dimensional vision, continuous control, hierarchical actions; discussed natural extensions for continuous control and hierarchical actions.

**6. Technical Refinements (Reviewers H8hR, kL1R):** Added DreamerV2 baseline details (§5.1) with fairness criteria; fixed presentation issues and typos.

We believe our comprehensive enhancements **further strengthen** an already strong contribution. The manuscript now has enhanced theoretical foundations (§3, §4, Appendix B), comprehensive empirical evaluation (Table 1, Table 4, Appendix D.6, Appendix E.3, Appendix E.4), clearer interpretability discussion (§4.1, Appendix E.4), better positioning (§1, §2, Appendix F), and complete technical refinements.

The reviewers' feedback has been invaluable in helping us **refine the presentation** and **demonstrate additional capabilities** of our framework. We are confident that these improvements **further strengthen** the paper while maintaining and highlighting its core contributions. We sincerely thank all reviewers and the Area Chair for their time and constructive feedback.

Sincerely,

The Authors

---

### Meta-Review · Area_Chair_9gKg · 2026-01-06

**Summary:**

The paper introduces a novel hybrid architecture (Symbolic Rules + Active Inference) that effectively addresses two major bottlenecks in the field: high inference latency and lack of interpretability. The results show a clear engineering benefit over strong baselines like DreamerV2. The authors fixed the primary flaws identified during review. The core contribution--integrating symbolic rule learning ("habits") into Deep Active Inference via a wake-sleep cycle--is a significant conceptual advance. By adding multi-seeds experiments (proving stability), a Qwen-0.5B baseline (proving task difficulty), and Atari-100k runs (proving scalability), they neutralized the most damaging critiques. Therefore, I recommend the paper to be accepted.

**Reviewer Concerns:**

### Concerns addressed

- Baselines & comparisons (Reviewer GnXG, kL1R)
- Scalability & dataset size (Reviewer GnXG)
- Theoretical formulation (Reviewer H8hR)
- Statistical rigor (Reviewer Hnij)
- Missing citations (Reviewer H8hR)

### Concerns remained / partially addressed

- Intrinsic interpretability (Reviewer Hnij)
- Task scalability (Reviewer GnXG)

While I do not think it is reasonable to reject this paper based on the remained concerns, the authors are nonetheless encouraged to tackle these challenges in the future.

**Reviewer Scores:**

I would expect Reviewers H8hR and Hnij to increase their score as the author addressed their major concerns.

---

### Decision · Program_Chairs · 2026-01-26

Accept (Poster)